# Backward Oversmoothing: why is it hard to train deep Graph Neural Networks?

**Nicolas Keriven** [1]

## Abstract

Oversmoothing has long been identified as a major limitation of Graph Neural Networks (GNNs): input node features are smoothed at each layer and converge to a constant non-informative representation, *if the weights of the GNN are sufficiently bounded*. This assumption is crucial: if, on the contrary, the weights are sufficiently large, then oversmoothing may be compensated. Theoretically, GNN could thus *learn* to not oversmooth. However, this does not really happen in practice, which prompts us to examine oversmoothing from an *optimization* point of view.

In this paper, we analyze *backward oversmoothing*, that is, the notion that backpropagated errors are also subject to oversmoothing from output to input. With non-linearities, we outline the key role of the *interaction* between forward and backward smoothing. Then, we show that, due to backward oversmoothing, GNNs provably exhibit many *spurious stationary points*: as soon as the *last* layer is trained, the *whole* GNN is at a stationary point. As a result, we can exhibit regions where gradients are near-zero while the loss stays high. Additionally, we prove that this is *specific* to GNNs, and does not necessarily hold for Multi-Layer Perceptrons. This paper is a step toward a more complete comprehension of the optimization landscape of GNNs.

## 1. Introduction

Graph Neural Networks (GNNs) (Scarselli et al., 2009; Bruna et al., 2014) are deep architectures that have *de facto* become the state-of-the-art for machine learning problems on graphs (Bronstein et al., 2021; Hu et al., 2020; Wu et al., 2022), with numerous applications ranging from chemistry

(Sypetkowski et al., 2024) to physics (Martínez et al., 2019), social networks analysis and epidemiology (Meirom et al., 2020), combinatorial optimization (Cappart et al., 2021), and many others. Given a graph of size $n$ represented by a graph-matrix $P \in \mathbb{R}^{n \times n}$ (that can be various flavors of normalized adjacency, Laplacian...) and input node features as rows of $X^{(0)} \in \mathbb{R}^{n \times d_0}$, a vanilla GNN follows the propagation equation:

$$X^{(k)} = \rho(PX^{(k-1)}W^{(k)}) \tag{1}$$

where $W^{(k)} \in \mathbb{R}^{d_{k-1} \times d_k}$ are learnable weights and $\rho$ is a non-linear activation function applied element-wise. A common interpretation is that GNNs realize "message-passing" through the matrix $P$. Many variants of message-passing exist, including some that cannot be written as fixed matrix multiplication (Veličković et al., 2018), but most classical GNNs follow this simple architecture.

While theoretical and empirical studies of GNNs form a large literature, with e.g. significant progress in the understanding of GNN expressivity or generalization (Vasileiou et al., 2025), some early topics and fundamental issues remain, to this day, active research questions (Morris et al., 2024). Among them, **oversmoothing** (Li et al., 2018; Oono & Suzuki, 2020; Rusch et al., 2023) is a long-standing problematic phenomenon. It is rather unique to GNNs, and easy to formulate: since node representations are mixed by the matrix $P$ ("*smoothed*") at each layer, when the GNN becomes very deep they converge to a constant, uninformative limit. This prevents regular GNNs from being too deep, which is an issue: it has been theoretically shown that GNN need depth for optimal performance (Loukas, 2020; Keriven, 2022), and shallow GNNs are subject to *underreaching*, where distant nodes do not communicate at all (Lu et al., 2023). Oversmoothing has been the topic of a large literature (Rusch et al., 2023), with many strategies to mitigate it, such as different flavors of skip connections (Li et al., 2021) or normalization (Zhao & Akoglu, 2020). Nevertheless, there are still some fundamental open questions about the effect of depth in GNNs in general.

Oversmoothing is rather easy to theoretically prove under appropriate hypotheses (Oono & Suzuki, 2020; Wu et al., 2023) (see Sec. 3.1). Typically, $P$ is assumed to be a stochastic matrix[1] that shrinks all directions but the constant one.

---
[*]Equal contribution [1]CNRS, IRISA, Université de Rennes, Rennes, France. Correspondence to: Nicolas Keriven <nicolas.keriven@cnrs.fr>.

*Proceedings of the 43rd International Conference on Machine Learning*, Seoul, South Korea. PMLR 306, 2026. Copyright 2026 by the author(s).

---
[1]i.e. $P1_n = 1_n$. As we remark later, non-stochastic matri-

Therefore, *assuming that the weights $W^{(k)}$ do not expand too much the node representations' norm*, typical measures of oversmoothing (Rusch et al., 2023) converge exponentially fast to zero as $k$ increases (see Sec. 3.1). The hypothesis on the weights is crucial: if they are large[2], then oversmoothing may not happen. As such, the GNN *could learn* to not oversmooth, as suggested by (Yang et al., 2020; Epping et al., 2024). However: a) it is not common to directly *initialize* expanding weights, as this can lead to explosion of the gradients, and b) when using regular initialization and gradient descent, vanilla deep GNNs generally do *not* learn to counteract oversmoothing, even though this would clearly lead to better minimizers of the loss. There is therefore a problem *with the optimization process itself*.

**Contribution.** In this paper, we thus look at oversmoothing from the optimization point of view. For this we put forth the notion of **backward oversmoothing**: just as the node representations are oversmoothed as $k$ increases (in the "forward"), then so are the backpropagated errors, when $k$ decreases from output to input. We show that, in the presence of a non-linear $\rho$, it is the **interaction** between forward smoothing and backward smoothing that eventually produces backward oversmoothing *at the middle layers* (Thm. 3.4 and Fig. 1, left). Then, our main result shows that backward oversmoothing produces many **spurious stationary points**: we prove that, as soon as *the last layer* is trained, then *the whole GNN* is in a near stationary-point (Thm. 4.2). We argue that this is the main reason why deep GNNs are "hard" to train: they tend to quickly get to an easy-to-reach local minima (Fig. 1, center), with high loss (Cor. 4.4). Moreover, we also show that regular Multi-Layer Perceptron (MLP) are *not* subjected to the same phenomenon (Prop. 4.5), making our result *specific* to GNNs. Our results rely on a precise characterization of the smoothed limit of the backward signal – which is generally not feasible in *forward* oversmoothing due to the non-linearity $\rho$, hence the plethora of studies that study oversmoothing only for linear GNNs (Keriven, 2022; Wu et al., 2023; Park & Kim, 2024; Chen et al., 2025; Peng et al., 2024). We show that, surprisingly, this characterization *is* possible in the backward, even with non-linear $\rho$.

Note that this paper does *not* propose new methods to mitigate oversmoothing: there are a great number of them already (Rusch et al., 2023) and it is slightly out of scope here. Similarly, we only examine vanilla GNNs here, and leave the analysis of existing strategies to mitigate oversmoothing for a future dedicated paper, as we expect it to be quite substantial (Chen et al., 2025). Our primary goal

ces lead to different limits, and deviate from the definitions of oversmoothing in (Rusch et al., 2023).

[2]A weight matrix whose spectral norm is larger than 1 will be said to be *expanding*.

is to explore important aspects of oversmoothing from an optimization point of view that were relatively ignored.

**Related work.** Theoretical optimization for deep learning is a vast field that is still riddled with open questions. Many existing results characterize the existence or absence of "bad" critical points that are local minima or saddle points (Yun et al., 2018; 2019; Zhou & Liang, 2018; Venturi et al., 2019; Ding et al., 2019). Other works study the convergence to these stationary points (Arora et al., 2019; Jin et al., 2019; Zhang et al., 2022). For both line of results, the situation can be reasonably well-characterized with linear activations, which is therefore the focus of many existing works (Yun et al., 2018; Arora et al., 2019). In the non-linear case, the existence of local minima is almost always guaranteed (Yun et al., 2019; Ding et al., 2019). In the specific case of GNNs, the authors in (Xu et al., 2021) proves global convergence of *linear* GNNs and study the effect of skip connections on convergence rates, while (Du et al., 2019) study the convergence of Graph Neural Tangent Kernels, which corresponds to infinitely-wide GNNs. Note that in general, these works *adapt* existing results from MLPs to GNNs, while in contrast, the results and intuitions presented in this paper *are specific to GNNs*, and do *not* apply to MLPs (see Thm. 4.2 and Prop. 4.5). Overall, optimization for GNNs remains a relatively under-explored topic from a theoretical point of view (Morris et al., 2024).

As mentioned above, oversmoothing is a long-standing problem in GNNs (Li et al., 2018; Oono & Suzuki, 2020), see (Rusch et al., 2023) for a survey. Typical strategies to mitigate it include variants of skip connections (Li et al., 2021; Pei et al., 2024; Chen et al., 2025) or normalization (Zhao & Akoglu, 2020). On the theoretical side, various measures of diversity of node representations have been proven to converge exponentially fast to zero (Oono & Suzuki, 2020; Wu et al., 2023) under some hypotheses, including in the attentional case (Wu et al., 2023). Skip connections have only recently been proven to indeed theoretically mitigate oversmoothing (Chen et al., 2025). On the other hand, depth is also provably useful for GNNs (Loukas, 2020; Keriven, 2022). Modern approaches to oversmoothing also relate it to *over-squashing* (Fesser & Weber, 2023; Álvaro Arroyo et al., 2025), where deep GNNs may lose information about distant neighborhoods if they are not wide enough. However we keep our focus on depth and oversmoothing here.

To our knowledge, there are not many works at the intersection of optimization and oversmoothing. Recently, in (Álvaro Arroyo et al., 2025), the authors relate oversmoothing to vanishing gradients, however: a) they assume Gaussian weights, and b) to our understanding, by assuming contracting Jacobians they place themselves in a case where *the forward signal itself converges to zero* with increasing layer index. This of course results in oversmoothing (a zero

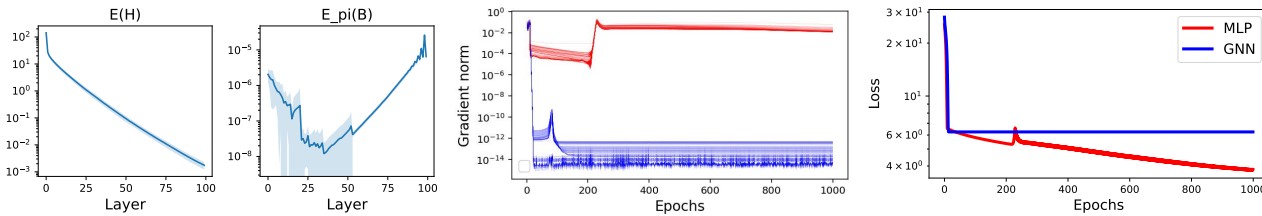

*Figure 1.* GNN with $P = D^{-1}A$ or MLP with 100 layers, on the WikiCS dataset (Mernyei & Cangea, 2020). **Left**: illustration of forward oversmoothing by measuring $\mathcal{E}(H^{(k)})$ (left) and backward oversmoothing by measuring $\mathcal{E}_\pi(B^{(k)})$ (right). **Center**: norms of gradients $\partial \mathcal{L}/\partial W^{(k)}$ for each layer wityh respect to epoch, for MLP (red) and GNN (blue). **Right**: corresponding losses. Other datasets are illustrated in App. B.

signal is of course constant), but we will rather look at a more general case where the forward signal can oversmooth without necessarily converging to 0 itself (as we show in Prop. 3.6). In (Peng et al., 2024), the authors do indeed argue that trainability is a more prominent problem than oversmoothing itself, however they again relate the problem to the traditional difficulty of training deep networks in the linear case, similar to (Álvaro Arroyo et al., 2025). Closer to us, Park & Kim (2024) introduce the idea of backward oversmoothing (calling it *gradient oversmoothing*) and show that the backpropagated error is smoothed from output to input. However, their work has important limitations: a) it is only valid in the linear case ($\rho = Id$), b) they do not explain why precisely having a smoothed backpropagated error is detrimental for optimization, and c) more importantly, at the time of this writing, there is a crucial error in their proof: they consider the symmetric GCN matrix $P = D^{-1/2}AD^{-1/2}$, which *does not lead to oversmoothing towards constant representation* (which they wrongly claim), as it is not stochastic, but toward the square root of the degree vector instead. This is not a detail: typical stochastic matrices $P$ are *not symmetric*, and, as we will see, this severely complexifies the proof of backward oversmoothing, as $P^\top$ plays a crucial role. In this paper, we do consider a general $\rho$, as well as stochastic non-symmetric $P$. We use backward oversmoothing to formally prove the existence of spurious stationary points, which we argue is the main reason why deep GNNs are hard to train.

**Outline.** We start with preliminaries in Sec. 2. In Sec. 3 we recall the classical forward oversmoothing, then show backward oversmoothing. In Sec. 4, we then prove our main result relating to stationary points for deep GNNs. We also discuss the difference between GNNs and MLPs. We conclude in Sec. 5. Proofs are provided in the appendix, as well as discussions and numerical illustrations.

## 2. Preliminaries

**Notations.** The norm $\|\cdot\| = \|\cdot\|_2$ is the euclidean norm for vectors and spectral norm (max singular value) for matrices.

The norm $\|\cdot\|_F$ is the Frobenius norm. The sign $\lesssim$ is an upper bound up to a multiplicative constant that does not involve the number of layers $L$. For $A$ and $B$ of the same size, $A \odot B$ is the Hadamard (element-wise) product.

**Stochastic message-passing matrix.** As described above, GNNs typically use matrices $P \in \mathbb{R}_+^{n \times n}$ to perform message-passing. It is well-known (Oono & Suzuki, 2020; Wu et al., 2023) that oversmoothing towards constant node representations as defined in (Rusch et al., 2023) appears for **stochastic** matrices, i.e. that satisfies $P1_n = 1_n$. On the contrary, as mentioned in the introduction, non-stochastic matrices may result in "oversmoothing" towards other limits, and therefore technically deviate from the definition of oversmoothing from (Rusch et al., 2023). We thus make the following assumption.

**Assumption 2.1** (Stochastic propagation matrix). The matrix $P \in \mathbb{R}_+^{n \times n}$ is stochastic and irreducible[3].

Replacing non-zero entries in $P$ by one, and viewing it as the adjacency matrix of a directed graph, $P$ is irreducible if and only if this directed graph is strongly connected. As Markov chain transition matrices, irreducible stochastic matrices have a unique stationary distribution $\pi \in \mathbb{R}_+^n$ with $\pi^\top 1_n = 1$ which is a left-eigenvector for the 1 eigenvalue: $\pi^\top P = \pi^\top$. Moreover, we have

$$P^k \xrightarrow[k\to\infty]{} 1_n\pi^\top \tag{2}$$

Typically, this convergence is exponentially fast, which we formulate in the following assumption.

**Assumption 2.2** (Convergence speed). There is a constant $C_P > 0$ and $0 < \lambda_P < 1$ such that for all $k > 0$,

$$\left\|P^k - 1_n\pi^\top\right\|_2 \leqslant C_P\lambda_P^k \tag{3}$$

When $P$ is diagonalizable $P = U^{-1}\Lambda U$ and 1 is a simple eigenvalue, this assumption is satisfied with $\lambda_P = |\lambda_2|$ the absolute value of the second largest eigenvalue of $P$ (which

---

[3]a matrix is irreducible if it cannot be permuted to a block triangular matrix

is indeed smaller than 1 by the Perron-Frobenius theorem) and $C_P = \|U\|_2 \|U^{-1}\|_2$. In the rest of the paper, we will also assume that $\pi_i > 0$ and denote by $\mu_\pi = \frac{\max(\pi_i)}{\min(\pi_i)}$.

A typical example of stochastic matrix for GNNs is the well-known random walk matrix $P = D^{-1}A$ where $D$ is the diagonal degree matrix. It corresponds to *mean aggregation* in message-passing, where each node representation is updated as the average of its neighbors. In this case, the stationary distribution is the normalized degree vector $\pi_i = \frac{d_i}{\sum_j d_j}$ where $d_i$ is the degree of node $i$, and $P$ is diagonalizable (and therefore satisfies Ass. 2.2) when the graph is strongly connected. Another illustrative example is the case where $P$ is symmetric, and therefore *bi*-stochastic. In this case $\pi = \frac{1_n}{n}$ is the uniform distribution, and computations are generally considerably simplified as $P$ is orthogonally diagonalizable. Bi-stochastic matrices are quite uncommon in GNNs, although they are for instance well-known in some domain such as distributed optimization (Boyd et al., 2006), where they are referred to as "gossip matrices". All the experiments in this paper are however performed with the more classical choice $P = D^{-1}A$.

**GNN: forward.**   Recall that the input node features are $X^{(0)} \in \mathbb{R}^{n \times d_0}$. We decompose the GNN (1) into the following notations:

$$F^{(k)} = PX^{(k-1)} \in \mathbb{R}^{n \times d_{k-1}}, \quad \text{(message-passing)}$$
$$H^{(k)} = F^{(k)}W^{(k)} \in \mathbb{R}^{n \times d_k} \quad \text{(weights multiplication)}$$
$$X^{(k)} = \rho(H^{(k)}), \quad \text{(activation function)}$$
$$out = H^{(L)} \in \mathbb{R}^{n \times d_L}, \quad \text{(output after } L \text{ layers)}$$

where $\rho$ is a non-linear activation function applied elementwise. We call $F^{(k)}$ the **forward signal**, going "from" input ($k = 0$) to output ($k = L$). We assume that the input node features are bounded $\left\|X^{(0)}_{i,:}\right\|_2 \leqslant D_{\mathcal{X}}$. The following assumption encompasses most activation functions.

**Assumption 2.3.** The function $\rho$ is 1-Lipschitz, and $|\rho(x)| \leqslant |x|$.

**Spectral norm of the weights.**   A crucial ingredients in oversmoothing is the "contracting" vs. "expanding" nature of the weights $W^{(k)}$. This is generally quantified by $s_k = \|W^{(k)}\|_2$ the leading singular value of $W^{(k)}$, or more simply the maximal $s := \max_k s_k$. Note that the $s_k$ evolve over the optimization process, however here most of our results are expressed at a particular point in time and involve the "current" $s_k$ without ambiguity.

**Loss function.**   We consider a loss function $\mathcal{L} : \mathbb{R}^{n \times d_L} \to \mathbb{R}_+$ over the graph nodes, decomposed as:

$$\mathcal{L}(H) = \frac{1}{n} \sum_{i=1}^n \ell_i(h_i) \tag{4}$$

for some functions $\ell_i : \mathbb{R}^{d_L} \to \mathbb{R}_+$, where the $h_i$ are the rows of $H$. Note that transductive semi-supervised learning can be emulated by having some $\ell_i = 0$, but we will generally bypass this detail here for simplification. We assume the following Lipschitz bound.

**Assumption 2.4.** There are constants $D_{\mathcal{L}}, D'_{\mathcal{L}} \geqslant 0$ such that $\left\|\frac{\partial \ell_i(h)}{\partial h}\right\|_2 \leqslant D_{\mathcal{L}} \|h\|_2 + D'_{\mathcal{L}}$.

Our two main running examples will be regression with the Mean Square Error (MSE) loss, or classification with the Cross-Entropy loss. We show the following in App. A.2.

**Example 1** (Regression). *In the regression case, the node labels $y_i \in \mathbb{R}^{d_L}$ are assumed bounded $\|y_i\| \leqslant 1$ (w.l.o.g.), and*

$$\ell_i(h) = \frac{1}{2} \|h - y_i\|^2 \tag{5}$$

*Assumption 2.4 is satisfied with $D_{\mathcal{L}} = D'_{\mathcal{L}} = 1$.*

**Example 2** (Classification). *In the classification case, the node labels are discrete $y_i \in \{1, \ldots, C\}$, the output dimension is $d_L = C$, and*

$$\ell_i(h) = -\log\left(e^{h_{y_i}} / \sum_c e^{h_c}\right) \tag{6}$$

*Assumption 2.4 is satisfied with $D_{\mathcal{L}} = 0$ and $D'_{\mathcal{L}} = C + 1$.*

**GNN: backward.**   First-order optimization methods rely on gradients $\frac{\partial \mathcal{L}}{\partial W^{(k)}}$, generally computed by backpropagation. For GNNs, the backpropagation equations are

$$\frac{\partial \mathcal{L}}{\partial W^{(k)}} = F^{(k)\top} B^{(k)}, \text{ where}$$
$$B^{(k)} := \frac{\partial \mathcal{L}}{\partial H^{(k)}} = \rho'(H^{(k)}) \odot \left(P^\top B^{(k+1)}(W^{(k+1)})^\top\right)$$
$$\text{and } B^{(L)} = \frac{\partial \mathcal{L}}{\partial H^{(L)}} \tag{7}$$

We have written these equations here to emphasize the object $B^{(k)} = \frac{\partial \mathcal{L}}{\partial H^{(k)}} \in \mathbb{R}^{n \times d_k}$ that we call the **backward signal**[4]. It satisfies a recursion equation, going "from" output ($k = L$) to input ($k = 0$). Often, the initialization $B^{(L)} = \frac{\partial \mathcal{L}}{\partial H^{(L)}}$ at the last layer represents the predictive "error" of the GNN: for instance, for the MSE loss, it is directly $B^{(L)} = \frac{1}{n}(H^{(L)} - Y)$ where $Y$ contains the labels $y_i$ as its rows. The full gradient $\frac{\partial \mathcal{L}}{\partial W^{(k)}} = F^{(k)\top} B^{(k)}$ is a product between the forward and the backward signal in the node dimension. Note that we do not perform mini-batching of nodes here[5] and only consider full gradients.

---

[4]It has several names in the literature, but here we emphasize its similarity to the forward signal.

[5]Indeed, batching is notoriously hard to define for node-tasks, as nodes are not independent. A GNN would typically require the $L$-hop subgraph around every batched node to be applied, which, for even moderate $L$, is generally the whole graph.

# 3. Oversmoothing: forward and backward

As described in the introduction, oversmoothing generally refers to the fact that node representations will converge to a constant representation as the number of layers increases, typically at exponential speed. Several metrics exist to measure the oversmoothing phenomenon, here we adopt the classical measure: for $X \in \mathbb{R}^{n \times d}$,

$$\mathcal{E}(X) := n^{-1/2} \min_{c \in \mathbb{R}^d} \left\| X - 1_n c^\top \right\|_F \quad (8)$$

which is the (normalized) distance from $X$ to the subspace $span(1_n)$. It is used in several landmark studies on oversmoothing (Oono & Suzuki, 2020), and satisfies the conditions formulated in (Rusch et al., 2023). More generally, we will denote the distance to $span(u)$, where $u \in \mathbb{R}^n$, by

$$\mathcal{E}_u(X) := n^{-1/2} \min_{c \in \mathbb{R}^d} \left\| X - u c^\top \right\|_F \quad (9)$$

such that $\mathcal{E}(\cdot) = \mathcal{E}_{1_n}(\cdot)$.

## 3.1. Forward oversmoothing

Oversmoothing can happen under several hypotheses. To be as generic as possible, similar to (Rusch et al., 2023) we formulate forward oversmoothing as a *definition*, and all our subsequent results will hold when it is satisfied. We may thus account for future results that might prove forward oversmoothing in more general cases.

**Definition 3.1.** We will say that **forward oversmoothing** holds with rate $0 < \lambda_f < 1$ and constant $C_f > 0$ if

$$\mathcal{E}(H^{(k)}) \leqslant C_f (\lambda_f s)^k \quad (10)$$

where we recall that $s = \max_k \left\| W^{(k)} \right\|_2$.

The following theorem gathers several known cases where forward oversmoothing holds, and is shown in App. A.1 for completeness.

**Theorem 3.2.** *Suppose that Assumptions 2.1 and 2.3 hold. If either:*

1. *$\rho = Id$ and Assumption 2.2 holds, in which case we consider $\lambda_f = \lambda_P$, or*

2. *there is $\lambda_f < 1$ such that*

$$\left\| J^\perp P J^\perp \right\|_2 \leqslant \lambda_f \quad (11)$$

*where $J^\perp := Id - 1_{n \times n}/n$ is the orthogonal projector onto $span(1_n)^\perp$.*

*Then forward oversmoothing holds with rate $\lambda_f$ and some constant $C_f$.*

In other words, either we assume the GNN to be linear, or we assume $P$ to contract every directions but the constant one. The linear case is very often adopted in the literature (Keriven, 2022; Wu et al., 2023; Park & Kim, 2024; Chen et al., 2025), as computations are greatly simplified and the constant limit can actually be characterized (see after). In this case, the mild Ass. 2.2 directly yields forward oversmoothing: we recall that it holds as soon as the graph is strongly connected, and $\lambda_P = |\lambda_2| < 1$ is the second-largest eigenvalue of $P$.

For non-linear $\rho$, the assumption (11) is, surprisingly, not immediate. **When $P$ is symmetric** and the eigenvalue 1 is simple, it is easy to see that (11) holds with $\lambda_f = |\lambda_2| < 1$, since $P$ is *orthogonally* diagonalizable. However, **when $P$ is non-symmetric**, it does *not* necessarily holds [6]. Hence, to our knowledge, *there is no theoretical proof of oversmoothing* in the "non-linear $\rho$, non-symmetric stochastic $P$" case. This is arguably an important missing piece in the literature, but it is not the purpose of the present paper. In practice however, forward oversmoothing seems to hold in the vast majority of cases: in App. B, we give several datasets for which $\left\| J^\perp P J^\perp \right\|_2 > 1$, but for which $\mathcal{E}(H^{(k)})$ seems to converge exponentially fast to 0, even with non-linear $\rho$.

From now on, we will thus *assume* that forward oversmoothing holds. We will also consider $\lambda = \max(\lambda_P, \lambda_f)$ and often express our bounds in term of $\lambda$ for simplicity.

**Expansion.** We see that the main condition for $\mathcal{E}(H^{(k)}) \to 0$ is $s\lambda < 1$. This authorizes *some* expansion $s > 1$, but $s$ is still bounded. In our results, we will put an emphasis on "how much" expansion is authorized, as we believe that this quantifies how much the weights should deviate from their initial value to escape the oversmoothing regime. More precisely, we will consider rates of the form $s \leqslant \lambda^{-\alpha}$ for some *expansion rate* $\alpha \leqslant 1$. As $\alpha$ decreases, this decreases the range of authorized $s$ above 1, up until $\alpha = 0$ and $s \leqslant 1$. As remarked in the introduction, if at some point during training $s > 1/\lambda$, then the GNN does not necessarily oversmooth (Epping et al., 2024). Hence the GNN could learn to *not* oversmooth (Yang et al., 2020), but again this does not really happen in practice, hence the need to examine the optimization process in more details.

**Limit signal.** Remark that while $X^{(k)}$ converges to a constant vector across nodes, due to the presence of the non-linear activation function $\rho$ it is generally difficult to explicitly characterize *what* this limit constant vector is (although some fixed point theorems may apply in certain circumstances (Álvaro Arroyo et al., 2025)). This is one of the reasons why many theoretical analyses of oversmoothing

---

[6]in particular, one may have $\|P\|_2 \geqslant 1$, even if all the eigenvalues of $P$ are less than 1. More precisely, $\|P\|_2 = 1$ if and only if $P$ is symmetric!

focus on the linear case $\rho = Id$, where the limit *can* be described with $P^k \to 1_n \pi^\top$ (Keriven, 2022; Wu et al., 2023; Park & Kim, 2024; Chen et al., 2025). As we will see later, this intuition plays a crucial role in backward oversmoothing, *even* for non-linear $\rho$.

### 3.2. Backward oversmoothing

Due to the multiplication by $P^\top$ from the output layer $k = L$ to the input $k = 1$, the backward signal will naturally also be oversmoothed. We need the following assumption.

**Assumption 3.3.** Assume that $\rho'$ is $D_\rho$-Lipschitz.

Note that this excludes ReLU (which is not fully differentiable anyway) but includes smoothed variants like softplus (Glorot et al., 2011). The following theorem is proved in App. A.3.

**Theorem 3.4.** *Assume that forward oversmoothing holds with rate $\lambda_f$ and constant $C_f$, and that Assumptions 2.1 to 3.3 hold. Then:*

$$\mathcal{E}_\pi(B^{(k)}) \leqslant C_L(\lambda_P s)^{L-k} + C_L' \lambda_f^k$$

*where $C_L = \frac{C_P A_L}{n}$, $C_L' = \frac{C_P C_f D_\rho \mu_\pi s^L A_L}{(1 - \lambda_P \lambda_f s)n}$ and $A_L = D_\mathcal{X} D_\mathcal{L} s^L + D_\mathcal{L}'$.*

We recall that $\mathcal{E}_\pi$ is the distance to $span(\pi)$, see (9). The bound on $\mathcal{E}_\pi(B^{(k)})$ involves two terms: one that decreases exponentially in $k$, the other one in $L - k$. Hence backward oversmoothing happens *at the middle layers* (Fig. 1, left), when *both* $k$ and $L - k$ are high. This is due to the term $\rho'(H^{(k)})$ in the backpropagation equations (7) which modulates the rows of $B^{(k)}$. *When the forward signal is oversmoothed* (high $k$), this modulation is almost constant, and backpropagation is exceedingly simple: $B^{(k)} \propto P^\top B^{(k+1)} W^{(k+1)^\top}$. It leads to backward oversmoothing after sufficiently many iterations (high $L - k$). On the contrary, at low $k$, the modulation $\rho'(H^{(k)})$ can be high. Hence, it is the *interaction* between forward and backward smoothing that leads to backward oversmoothing at the middle layers.

Theorem 3.4 describe a non-trivial relationship between $\lambda$ and $s$, making it difficult to assess the effects of the expansion rate $\alpha$. We can simplify it by instantiating the loss.

**Corollary 3.5.** *Take $q = 2$ for a regression problem with the MSE loss, or $q = 1$ for a classification problem with the cross-entropy loss. Consider that the assumptions of Theorem 3.4 hold, denote $\lambda = \max(\lambda_P, \lambda_f)$. If $s \leqslant \lambda^{-\alpha}$ where $0 \leqslant \alpha < 1 - \sqrt{1 - \frac{1}{q}}$, then for all $q\alpha < \beta < \frac{1-q\alpha}{1-\alpha}$,*

$$\mathcal{E}_\pi\left(B^{(\beta L)}\right) \lesssim \lambda^{(\beta - q\alpha)L} + \lambda^{(1 - q\alpha - (1-\alpha)\beta)L} \xrightarrow[L\to\infty]{} 0$$

In other words, taking the layer index $k = \beta L$ proportional to the depth $L$, with sufficiently low $\alpha$ we obtain an explicit exponential rate.

**Backward oversmoothing is not (necessarily) vanishing gradients.** While one might feel that backward oversmoothing is detrimental for GNN training, it is not clear exactly why this is true. A first idea would be to relate backward oversmoothing to *vanishing gradients*: if $\mathcal{E}_\pi(B^{(k)})$ vanishes, then maybe the *norm* $\left\|B^{(k)}\right\|_F$ vanishes altogether, as was shown in (Álvaro Arroyo et al., 2025) in very specific cases. This would be the very well-known phenomenon of vanishing gradient. However, this is not true in general, and we prove the following in App. A.5.

**Proposition 3.6.** *There is a graph that satisfies Ass. 2.1 with a symmetric $P$ with $\lambda_2 < 1$ such that the following holds. For all $L$, there is a GNN of depth $L$ that satisfies all the assumptions of Thm. 3.4 with $s \leqslant 1$ such that for all $k$:*

$$\left\|\partial\mathcal{L}/\partial W^{(k)}\right\|_F = 1 \qquad (12)$$

That is, we can have both backward oversmoothing (by Thm. 3.4) but with arbitrarily high gradients. Hence the answer is *not* vanishing gradients, at least not automatically. In the next section, we relate backward oversmoothing to *(spurious) stationary points*.

## 4. Backward oversmoothing and spurious stationary points

While backward oversmoothing is not directly responsible for vanishing gradients, in practice we observe that deep GNNs tend to quickly and easily get stuck in local minima (Fig. 1, center). This prompts us to analyze the stationary points of deep GNNs. While stationary points are not inherently bad, we will identify such points that we loosely call "spurious", in the sense that: they may incur arbitrarily high loss (Cor. 4.4), and they are entirely due to backward oversmoothing (Thm. 4.2 and 4.3) and *do not occur* for MLPs (Prop. 4.5). We thus argue that they represent a major difficulty in training deep GNNs.

We first introduce the following definition of near-stationary points (Shamir, 2020).

**Definition 4.1** ($\delta$-stationary point). The weight $W^{(k)}$ is said to be at a $\delta$-**stationary point** if

$$\left\|\partial\mathcal{L}/\partial W^{(k)}\right\|_F \leqslant \delta \qquad (13)$$

If this is true for all $1 \leqslant k \leqslant L$, then the GNN is said to be at a **global $\delta$-stationary point**.

Note that, in terms of vocabulary, we somewhat distinguish the notion of stationary points with that of vanishing gradient, even for small $\delta$. Indeed, vanishing gradients often

refers to the fact that gradient may have vastly *different* amplitudes across layers, and is generally understood to be a strictly detrimental phenomenon. On the contrary, stationary points can be detrimental – e.g., "bad" local minima, saddle points – or simply the sign of successful training: after all, characterizing the rates of convergence to near-stationary points is the main goal of non-convex optimization (Shamir, 2020; Hollender & Zampetakis, 2024).

We now turn to the main result of this paper. For convenience we define the function

$$\xi_q(\alpha) = \frac{1-(2q+1)\alpha+q\alpha^2}{2(1-\alpha)} \qquad (14)$$

and note that $\xi_q(\alpha) > 0$ for $\alpha \leqslant 1 + \frac{1}{2q} - \sqrt{1 + \frac{1}{4q^2}}$, which is roughly equal to 0.38 for $q = 1$ and 0.21 for $q = 2$. The following theorem is proved in App. A.6.

**Theorem 4.2** (A stationary output layer is a global stationary point). *Suppose that Assumptions 2.1 to 3.3 hold, and that forward oversmoothing holds with rate $\lambda_f$. Denote $\lambda = \max(\lambda_P, \lambda_f)$. Suppose that $s \leqslant \lambda^{-\alpha}$, with $\xi_q(\alpha) > 0$. Assume that the output forward signal is not zero: $\frac{1}{\sqrt{n}} \left\| F^{(L)} \right\|_F \geqslant D_F$, and that **the parameter $W^{(L)}$ is at a $\delta_{W_L}$-stationary point**. For $\delta > 0$, if*

$$L \gtrsim \frac{\log(1/(D_F\delta))}{\xi_q(\alpha)\log(1/\lambda)} \quad \text{and} \quad \delta_{W_L} \lesssim \lambda^{\alpha L} D_F \delta \quad (15)$$

*then **the GNN is at a global $\delta$-stationary point**.*

This theorem shows that, for sufficiently deep GNNs, under some mild assumptions, **every $\delta_{W_L}$-stationary point *for the output layer $L$ is also a *global $\delta$-stationary point**, for some small $\delta$ related to $\delta_{W_L}$ and $L$. In other words, as soon as the last layer of the GNN is "trained", gradients vanish *at every layers* and the GNN hardly trains anymore. As one might guess, the last (linear) layer is exceedingly easy to train, which results in many easy-to-reach stationary points. Note that, while backward oversmoothing is the underlying mechanism for this result as we will explain next, $\delta$-stationarity indeed happens *at every layers*.

**Sketch of proof.** We now provide a sketch of proof of Theorem 4.2, as we consider that it provides interesting intuitions. The full proof can be found in App. A.6.

In forward oversmoothing, the signal $X^{(k)}$ converges to a constant node representation when $k$ increases. As mentioned in the previous section, in the linear case $\rho = Id$ it is easy to characterize the limit by leveraging $P^k \rightarrow 1_n\pi^\top$, which is not possible anymore in the non-linear case. The situation is different for backward oversmoothing. As mentioned earlier, *when the forward signal is oversmoothed* (near the output layer), the term $\rho'(H^{(k)})$ is approximately constant, and **the update equation is approximately** $B^{(k)} \propto P^\top B^{(k+1)}(W^{(k+1)})^\top$, which is nothing

more than a linear GNN! Hence we *can* approximate the limit representation of the backward oversmoothed signal, and it relates to the *average* of the output error: at the middle layers, $B^{(k)} \propto \pi 1_n^\top B^{(L)}(W_L \dots W_{k+1})^\top$, using $(P^\top)^k \rightarrow \pi 1_n^\top$. We thus obtain the following theorem, which is the core of the proof of Thm 4.2. A more detailed version, and a proof, can be found in App. A.9.

**Theorem 4.3.** *Take $q = 2$ for a regression problem, or $q = 1$ for a classification problem. Suppose that Assumptions 2.1 to 3.3 hold, and that forward oversmoothing holds with rate $\lambda_f$. Denote $\lambda = \max(\lambda_P, \lambda_f)$. If $s \leqslant \lambda^{-\alpha}$ with $\xi_q(\alpha) > 0$, the GNN is at a global $\delta$-stationary point, with*

$$\delta \lesssim \lambda^{\xi_q(\alpha)L} + \lambda^{-\alpha L} \left\| 1_n^\top B^{(L)} \right\|_2 \qquad (16)$$

The key component here is the norm of $1_n^\top B^{(L)} = \frac{1}{n}\sum_i \frac{\partial \ell_i(h_i)}{\partial h_i}$, which is indeed the *average* of the output error of the GNN. One notable aspect of this result is that it applies *to every layer $k$*, even though oversmoothing is the underlying phenomenon – first, middle and last layers are indeed treated differently in the proof. To conclude with the proof of Theorem 4.2, we then bound the average output error $\left\| 1_n^\top B^{(L)} \right\|_2$ when the last layer is at a stationary point: *since the output of the GNN is almost constant due to forward oversmoothing*, when the last layer is trained, the *average* error automatically vanishes.

Note that the assumption that $\frac{1}{\sqrt{n}} \left\| F^{(L)} \right\|_F \geqslant D_F$ is satisfied when $F_{i,:}^{(L)}$ is bounded away from 0 for each node, but it is somewhat a proof artifact. Indeed, since $\partial\mathcal{L}/\partial W^{(L)} = F^{(L)\top} B^{(L)}$, one may get to $\delta_{W_L}$-stationarity simply because $F^{(L)} \approx 0$ and not because of any training of $W_L$, which is not what we want here. If $F^{(L)} \approx 0$, it would probably be possible to adapt the theorem to replace the $\delta_{W_L}$-stationarity of $W^{(L)}$ to that of the first non-zero layer $F^{(k)}$, but this would be exceedingly verbose and does not conceptually change the result.

**Spurious stationary point.** As mentioned in the introduction, stationary points are not inherently bad: reaching a stationary point *is* the goal of optimization (Shamir, 2020; Hollender & Zampetakis, 2024), in the hope that it is a good (local) minimum in the non-convex case. A further question is, can we leverage Theorem 4.2 or 4.3 to show that deep GNNs can have a global $\delta$-stationary point, for an arbitrary *high* loss? Such a result would be quite unusual in non-convex optimization, as the existence of stationary point is generally decorrelated from the value of the loss in itself. We give an example below, proved in App. A.7.

**Corollary 4.4** (Spurious stationary points). *Suppose that Assumptions 2.1 to 3.3 hold, and that forward oversmoothing holds with rate $\lambda_f$. Denote $\lambda = \max(\lambda_P, \lambda_f)$. Consider*

*the regression case, and assume that the labels are centered:*

$$\sum_i y_i = 0$$

*Denote by $\sigma_y^2 = \frac{1}{n} \sum_i \|y_i\|_2^2$. Suppose that $s \leqslant \lambda^{-\alpha}$, with $\xi_2(\alpha) > 0$, and the last weight is such that $\|W_L\|_2 \leqslant \varepsilon_{W_L}$. For $\delta, \varepsilon > 0$, if*

$$L \gtrsim \frac{\log(1/\delta)}{\xi_2(\alpha)\log(1/\lambda)} \text{ and } \varepsilon_{W_L} \lesssim \min\left(\lambda^{2\alpha L}\delta, \frac{\lambda^{\alpha L}}{\sigma_y}\varepsilon\right)$$

*then the GNN is at a **global $\delta$-stationary point**, and **the loss satisfies***

$$\mathcal{L}(H^{(L)}) \geqslant \sigma_y^2/2 - \varepsilon \tag{17}$$

In other words, for a centered regression problem and sufficiently deep GNNs, it is easy to find a flat region of near-stationary points such that *the loss is uniformly high*, close to the variance of the labels. While we constructed this particular example around $W_L = 0$, it is likely that many other flat regions with high loss could be constructed on the same principles, by leveraging Thm. 4.3.

**GNNs are not MLPs.** It is known, both in theory and in practice, that the landscape of MLPs is riddled with local minima and saddle points (Venturi et al., 2019; Zhou & Liang, 2018; Ding et al., 2019; Yun et al., 2019). Can we make certain that the stationary points identified by Thm. 4.2 are indeed *specific* to GNNs? The following proposition answers by the affirmative.

**Proposition 4.5** (MLPs are not GNNs). *Consider a regression problem, in the MLP case $P = Id$. For all $n$ there is a data distribution such that the following holds. For all depth $L > 0$, there is an MLP that satisfies Assumptions 2.3 and 3.3, such that $s \leqslant 1$, $\frac{1}{\sqrt{n}}\|F^{(L)}\| = 1$, $\frac{\partial\mathcal{L}}{\partial W^{(L)}} = 0$, but there exists $k$ such that*

$$\left\|\frac{\partial\mathcal{L}}{\partial W^{(k)}}\right\| = 1 \tag{18}$$

This proposition indeed contradicts Thm. 4.2 for MLPs: there is a regression problem and an arbitrarily deep MLP that satisfy all the Assumptions of Thm. 4.2 – including that $F^{(L)}$ is lower-bounded and $W_L$ is at a stationary point – such that at least one gradient does not vanish. Hence, our results are not another characterization of existing stationary points in deep learning, but are instead *specific* to GNNs.

**Mitigating oversmoothing.** Finally, we recall that the results presented in this paper only apply to vanilla GNNs, without the common fixes to oversmoothing: skip connections, normalization, and so on. We leave the detailed analysis of these methods for future work, as we expect it to be quite substantial (similar to (Chen et al., 2025)) but state

the following brief remarks. Intuitively, skip connections in the forward $X^{(k)} = X^{(k-1)} + \rho(PX^{(k-1)}W^{(k)})$ also results in skip connections in the backward signal. Hence, we may expect to extend existing analyses that skip connections provably mitigate forward oversmoothing (Chen et al., 2025) to the backward signal. One known difficulty is that, done naively, skip connections can result in exploding signal amplitude: since the latter may more or less scale as $(1+s)^k$, one must generally choose to initialize the weights with $s = O(1/k)$, as is sometimes done in ResNets (Zhang et al., 2019). Intuitively, this is also true for the backward signal in GNNs (Park & Kim, 2024). Our analysis of expanding weights with rate $s \leqslant \lambda^{-\alpha}$ would therefore be significantly different. Concerning normalization strategies, they are probably much harder to analyze, as highly non-linear operations. In the future, we may approach them for toy examples, or random weights with particular distributions (Chen et al., 2025).

**Numerical illustrations.** In App. B, we provide a few numerical illustrations of our results on different synthetic and real datasets, similar to Fig. 1. In particular, we observe that forward oversmoothing seem to always hold, even when (11) is not satisfied and $\rho$ is non-linear. We also briefly illustrate the effects of skip connections, even though this is not covered by the theory in this paper. The code is available in the supplementary material.

## 5. Conclusion

In this paper, we explored the links between oversmoothing and optimization. Our major finding is that deep GNNs possess many *spurious near-stationary points*, that are a) easy-to-reach: as soon as the last layer is stationary; and b) fully specific to GNNs: we exhibit a counter-example MLP. The underlying mechanism is *backward oversmoothing*, that is, the backward errors are also prone to oversmoothing. Key observations are that: a) for non-linear GNNs, backward oversmoothing *interacts* in non-trivial ways with forward oversmoothing, and b) when the forward signal is also oversmoothed, the recursive iterations on the backward signal are almost linear, meaning that we can fully *characterize the constant oversmoothed limit*, despite the non-linearity $\rho$. The latter observation is the key to relating stationarity of the GNN with the *average output error*. Our results hold even in the general case with non-linear $\rho$ and non-symmetric $P$.

**Outlooks.** As we have seen, forward oversmoothing is not entirely solved, and this is an important missing piece in the literature, even if it is not directly related to the results presented in this paper.

The most direct extension of this paper would be to cover the mitigation strategies for oversmoothing, skip connection

and normalization. Theoretical analysis of skip connections are highly non-trivial (Chen et al., 2025), but might be adapted in the backward case. On the practical side, our results may help design better *optimization algorithms* for GNNs, e.g. not based on vanilla backpropagation (Zhao et al., 2024). Another direct extension would be to analyze attention-based models, for which it is known that forward oversmoothing holds in the linear case (Wu et al., 2023), but for which the backward equations are significantly different.

More generally, given the relative scarcity of theoretical works on optimization for GNNs (Morris et al., 2024), our results are a step towards a better comprehension of this field. In particular, we have not analyzed the *dynamics* of optimization algorithms, which would be necessary to prove that actual algorithms do indeed reach the "spurious" stationary points that we identified in this paper.

Finally, as mentioned above, a major characteristic of our results and intuitions is that *they are specific to GNNs*, which is not so common – a large part of the literature adapt existing results from MLP to GNNs. An important path for future work would thus be to identify more fundamental differences between existing results for MLPs (Yun et al., 2018; 2019; Zhou & Liang, 2018; Venturi et al., 2019; Ding et al., 2019) and GNNs, beyond simple adaptations.

## Impact Statement

This paper is mostly theoretical in nature, and presents work whose goal is to advance the field of Machine Learning. There are potential societal consequences of our work, however due to the theoretical nature of the work, these impacts are mostly indirect, none which we feel must be specifically highlighted here.

## Acknowledgements

This work was funded by the European Union ERC-2024-STG-101163069 MALAGA.

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

# A. Proofs

For $X \in \mathbb{R}^{n \times d}$, we define $\|X\|_{\infty,2} := \max_i \|X_{i,:}\|$. We also denote $c_\pi = \|\pi\|_2$, which is in $O(1/\sqrt{n})$ when $\mu_\pi = O(1)$.

## A.1. Proof of Thm. 3.2

By Lemma C.1, we have that $\mathcal{E}(H) = \left\|J^\perp H\right\|_F / \sqrt{n}$.

**Case 1.** If $\rho = Id$, then

$$\left\|J^\perp H^{(k)}\right\|_F = \left\|P^k X^{(0)} W^{(1)} \dots W^{(k)}\right\|_F \leqslant \left\|J^\perp P^k X^{(0)}\right\| s^k$$

using $\|AB\|_F \leqslant \|A\|_F \|B\|_2$. Then, since $\left\|J^\perp P^k JX\right\| = \left\|J^\perp JX\right\| = 0$ and $\left\|J^\perp 1_n \pi^\top X\right\| = 0$ and $\left\|J^\perp\right\|_2 = 1$ (it's a projector), we have

$$
\begin{aligned}
\left\|J^\perp P^k X^{(0)}\right\|_F &= \left\|J^\perp P^k J^\perp X^{(0)}\right\|_F = \left\|J^\perp (P^k - 1_n \pi^\top) J^\perp X^{(0)}\right\|_F \\
&\leqslant \left\|J^\perp\right\|_2 \left\|(P^k - 1_n \pi^\top) J^\perp X^{(0)}\right\|_F \\
&\leqslant \left\|(P^k - 1_n \pi^\top) J^\perp X^{(0)}\right\|_F \\
&\leqslant \left\|P^k - 1_n \pi^\top\right\|_2 \left\|J^\perp X^{(0)}\right\|_F \\
&\leqslant \left\|J^\perp X^{(0)}\right\|_F C_P \lambda_P^k
\end{aligned}
\tag{19}
$$

Hence forward oversmoothing holds with $\lambda_f = \lambda_P$ and $C_f = \mathcal{E}(X^{(0)})C_P$.

**Case 2.** By recursion, we have: first, as in the previous case,

$$\mathcal{E}(XW) \leqslant \mathcal{E}(X)\|W\|_2 \tag{20}$$

Then, we remark that $\mathcal{E}$ can be reformulated as a sum over pairwise differences: by a simple computation,

$$\mathcal{E}(X)^2 = \frac{1}{2n^2} \sum_{i,j} \|X_{i,:} - X_{j,:}\|^2$$

From this expression, since $\rho$ is 1-Lipschitz, we immediately get $\mathcal{E}(\rho(X)) \leqslant \mathcal{E}(X)$.

Finally, as in the previous case $\left\|J^\perp PJX\right\| = 0$ and thus

$$\left\|J^\perp PX\right\|_F = \left\|J^\perp PJ^\perp X\right\|_F \leqslant \left\|J^\perp P(J^\perp)^2 X\right\|_F \leqslant \lambda_f \left\|J^\perp X\right\|_F \tag{21}$$

since $(J^\perp)^2 = J^\perp$ and $\left\|J^\perp PJ^\perp\right\|_2 \leqslant \lambda_f$ by hypothesis. Hence

$$\mathcal{E}(PX) \leqslant \lambda \mathcal{E}(X)$$

Combining all these results

$$\mathcal{E}(H^{(k)}) = \mathcal{E}(P\rho(H^{(k-1)})W^{(k)}) \leqslant \lambda s \mathcal{E}(H^{(k-1)})$$

and an easy recursion shows that forward oversmoothing holds with rate $\lambda_f$ and constant $C_f = \mathcal{E}(X^{(0)})$.

## A.2. Assumption 2.4 for MSE and cross-entropy

For the MSE, we have

$$\frac{\partial \ell_i}{\partial h} = h - y_i \tag{22}$$

hence $D_\mathcal{L} = D'_\mathcal{L} = 1$ (since $y_i \leqslant 1$).

For the cross entropy loss with $C$ classes, we have

$$\left\|\frac{\partial \ell_i}{\partial h}\right\| \leqslant \left|\frac{\exp(h_{y_i})}{\sum_c \exp(h_c)} - 1\right| + \sum_{e \neq y_i}\left|\frac{\exp(h_e)}{\sum_c \exp(h_c)}\right| \leqslant C + 1 \tag{23}$$

hence $D_{\mathcal{L}} = 0$ and $D'_{\mathcal{L}} = C + 1$.

### A.3. Proof of Thm 3.4

Denote $\rho_k = \rho'(H^{(k)})$, $v_k = \rho_k^\top 1_n / n$, $V^{(k)} = W^{(k+1)\top}\mathrm{diag}(v_k)$, and $M_k = (J^\perp \rho_k) \odot \left(P^\top B^{(k+1)}W^{(k+1)\top}\right)$.

Remark that, since $\|v_k\|_\infty \leqslant 1$, for any $x$ such that $\|x\| = 1$,

$$\left\|V^{(k)}x\right\| = \left\|W^{(k+1)}(v_k \odot x)\right\| \leqslant s_{k+1}\|v_k\|_\infty\|x\| \leqslant s_{k+1}$$

such that $\left\|V^{(k)}\right\|_2 \leqslant s_{k+1}$.

Using $(1v^\top) \odot B = B \cdot \mathrm{diag}(v)$, we decompose $B^{(k)}$ as:

$$\begin{aligned}
B^{(k)} &= \rho_k \odot \left(P^\top B^{(k+1)}W^{(k+1)\top}\right) = (J\rho_k) \odot \left(P^\top B^{(k+1)}W^{(k+1)\top}\right) + M_k \\
&= P^\top B^{(k+1)}V^{(k)} + M_k
\end{aligned} \tag{24}$$

By a simple recursion,

$$B^{(k)} = \underbrace{(P^\top)^{L-k}B^{(L)}V^{(L-1)}\dots V^{(k)}}_{B_1 = B_1^{(k)}} + \underbrace{\sum_{\ell=k+1}^{L-1}(P^\top)^{\ell-k}M_\ell V^{(\ell-1)}\dots V^{(k)} + M_k}_{B_2 = B_2^{(k)}} \tag{25}$$

Since $\mathcal{E}_u(X + Y) \leqslant \mathcal{E}_u(X) + \mathcal{E}_u(Y)$ by Lemma C.1, we bound $\mathcal{E}_\pi$ for each term. For the first, using Lemma C.1 and Assumption 2.2,

$$\begin{aligned}
\mathcal{E}_\pi(B_1) &= \mathcal{E}_\pi\left((P^\top)^{L-k}B^{(L)}V^{(L-1)}\dots V^{(k)}\right) \leqslant \mathcal{E}_\pi\left((P^\top)^{L-k}B^{(L)}\right)s^{L-k} \\
&= \mathcal{E}_\pi\left(\left((P^\top)^{L-k} - \pi 1_n^\top\right)B^{(L)}\right)s^{L-k} \\
&\leqslant n^{-1/2}\left\|\left((P^\top)^{L-k} - \pi 1_n^\top\right)B^{(L)}\right\|_F s^{L-k} \\
&\leqslant n^{-1/2}\left\|P^{L-k} - 1_n\pi^\top\right\|_2\left\|B^{(L)}\right\|_F s^{L-k} \\
&\leqslant C_P(s\lambda)^{L-k}\left\|B^{(L)}\right\|_{\infty,2} \leqslant C_P(s\lambda)^{L-k}\frac{A_L}{n}
\end{aligned}$$

by Lemma C.4, using $\|\cdot\|_F \leqslant n^{1/2}\|\cdot\|_{\infty,2}$, where $A_L = s^L D_\mathcal{X}D_\mathcal{L} + D'_\mathcal{L}$.

For the second term,

$$\begin{aligned}
\mathcal{E}_\pi(B_2) &\leqslant \sum_{\ell=k+1}^{L-1}\mathcal{E}_\pi\left((P^\top)^{\ell-k}M_\ell V^{(\ell-1)}\dots V^{(k)}\right) + \mathcal{E}_\pi(M_k) \\
&\leqslant \sum_{\ell=k}^{L-1}\mathcal{E}_\pi\left((P^\top)^{\ell-k}M_\ell\right)s^{\ell-k} \\
&\leqslant \sum_{\ell=k}^{L-1}\mathcal{E}_\pi\left(\left((P^\top)^{\ell-k} - \pi 1_n^\top\right)M_\ell\right)s^{\ell-k} \\
&\leqslant n^{-1/2}C_P\sum_{\ell=k}^{L-1}(\lambda s)^{\ell-k}\|M_\ell\|_F
\end{aligned}$$

and then

$$\|M_k\|_F = \left\|(J^\perp \rho_k) \odot \left(P^\top B^{(k+1)} W^{(k+1)\top}\right)\right\|_F$$
$$\leqslant \|J\rho_k\|_F \left\|P^\top B^{(k+1)} W^{(k+1)\top}\right\|_\infty = n^{1/2}\mathcal{E}(\rho_k) \left\|P^\top B^{(k+1)} W^{(k+1)\top}\right\|_\infty$$

By Lemma C.3 and C.4, since $D_\pi^{-1} P^\top D_\pi$ is a stochastic matrix, we have

$$\left\|P^\top B^{(k+1)}(W^{(k+1)})^\top\right\|_\infty \leqslant \left\|P^\top B^{(k+1)}(W^{(k+1)})^\top\right\|_{\infty,2}$$
$$\leqslant \pi_{\max} \left\|D_\pi^{-1} P^\top D_\pi D_\pi^{-1} B^{(k+1)}(W^{(k+1)})^\top\right\|_{\infty,2}$$
$$\leqslant \pi_{\max} s \left\|D_\pi^{-1} B^{(k+1)}\right\|_{\infty,2} \leqslant \frac{\mu_\pi}{n} s^{L-k} A_L$$

and since $\rho'$ is $D_\rho$-Lipschitz,

$$\mathcal{E}(\rho_k) \leqslant D_\rho \mathcal{E}(H^{(k)}) \leqslant D_\rho C_f s^k \lambda_f^k$$

Hence

$$\|M_k\|_F \leqslant \frac{D_\rho C_f \mu_\pi s^L A_L}{\sqrt{n}} \lambda_f^k \tag{26}$$

Thus,

$$\mathcal{E}_\pi(B_2) \leqslant C_P \frac{\mu_\pi}{n} s^L A_L C_f D_\rho \sum_{\ell=k}^{L-1}(\lambda s)^{\ell-k} \lambda_f^k$$
$$\leqslant \frac{C_P C_f D_\rho \mu_\pi s^L A_L}{n} \lambda_f^k \sum_{\ell=k}^{L-1}(\lambda \lambda_f s)^{\ell-k}$$
$$\leqslant \frac{C_P C_f D_\rho \mu_\pi s^L A_L}{(1-\lambda\lambda_f s)n} \lambda_f^k$$

which concludes the proof.

### A.4. Proof of Corollary 3.5

In the regression case, we have $D_\mathcal{L} = 1$, while in the classification case, $D_\mathcal{L} = 0$. Hence, taking $\lambda = \max(\lambda_P, \lambda_f)$ and $s \leqslant \lambda^{-\alpha}$ with $0 \leqslant \alpha < 1$, the result of Thm 3.4 reads

$$\mathcal{E}_\pi(B^{(k)}) \lesssim \lambda^{L-k-\alpha(L-k)-(q-1)\alpha L} + \lambda^{k-q\alpha L}$$

where $q = 2$ in the regression case and $q = 1$ in the classification case, and the multiplicative constant does not depend on $L$. Choosing $k = \beta L$, we get

$$\mathcal{E}_\pi(B^{(k)}) \lesssim \lambda^{(\beta-q\alpha)L} + \lambda^{(1-q\alpha-(1-\alpha)\beta)L}$$

This goes to 0 as $L$ increases when $\beta > q\alpha$ and $\beta < \frac{1-q\alpha}{1-\alpha}$. This is possible when $q\alpha < \frac{1-q\alpha}{1-\alpha}$, which after a few computations leads to the condition $\alpha < 1 - \sqrt{1-1/q}$.

### A.5. Proof of Prop 3.6

Consider a trivial GNN with $\rho = Id$ and $W^{(k)} = 1$, with $d_k = 1$ for all layer $k$, a constant input signal $X^{(0)} = 1_n$, constant zero labels $Y = 0_n$, and the MSE loss $\mathcal{L}(H) = \frac{1}{2n}\|H - Y\|_2^2$. Then, since $P1 = 1$, $F^{(k)} = 1_n$ and $B^{(k)} = \frac{2}{n}1_n$ for all $k$. The gradient is therefore constant, equal to

$$\left|F^{(k)\top} B^{(k)}\right| = 1$$

## A.6. Proof of Thm 4.2

Our strategy is to bound $\varepsilon_n := \left\| 1_n^\top B^{(L)} \right\|_2$ and use Thm 4.3. We have

$$
\begin{aligned}
\delta_{W_L} &\geqslant \left\| \frac{\partial \mathcal{L}}{\partial H^{(L)}} \right\|_F = \left\| F^{(L)^\top} B^{(L)} \right\|_F \\
&\geqslant \left\| J F^{(L)^\top} B^{(L)} \right\|_F - \left\| J^\perp F^{(L)^\top} B^{(L)} \right\|_F \\
&\geqslant \left\| v 1_n^\top B^{(L)} \right\|_F - n \left\| B^{(L)} \right\|_{\infty,2} \mathcal{E}(F^{(L)}) \\
&\geqslant \|v\|_2 \, \varepsilon_n - A_L C_P \lambda C_f (s\lambda)^{L-1} D_\mathcal{X}
\end{aligned}
\tag{27}
$$

where $v = \frac{1}{n} 1_n^\top F^{(L)}$, since $\mathcal{E}(PX) \leqslant C_P \lambda \mathcal{E}(X)$ similarly to (19).

Since we have $\frac{1}{\sqrt{n}} \left\| F^{(L)} \right\|_F \geqslant D_F$ by hypothesis, we have

$$
\begin{aligned}
\|v\|_2 &= \frac{1}{\sqrt{n}} \|v 1_n\|_F = \frac{1}{\sqrt{n}} \left\| J F^{(L)} \right\|_F \\
&\geqslant \frac{1}{\sqrt{n}} \left\| F^{(L)} \right\|_F - \frac{1}{\sqrt{n}} \left\| J^\perp F^{(L)} \right\|_F \geqslant D_F - \mathcal{E}(F^{(L)}) \geqslant D_F - D_\mathcal{X} C_P \lambda C_f (s\lambda)^{L-1}
\end{aligned}
$$

Therefore, by combining the two, for $L$ large enough:

$$
\varepsilon_n \leqslant \frac{\delta_{W_L} + A_L C_P \lambda C_f (s\lambda)^{L-1} D_\mathcal{X}}{D_F - D_\mathcal{X} C_P \lambda C_f (s\lambda)^{L-1}}
$$

Combined with Thm 4.3, the GNN is at a global $\delta$-stationary point, with

$$
\delta \lesssim \frac{1}{D_F} \left( \lambda^{\xi_q(\alpha)L} + \lambda^{(1-(q+1)\alpha)L} + \lambda^{-\alpha L} \delta_{W_L} \right)
$$

Equating the two terms and inverting we get conditions on $L$ and $\bar{\delta}$ to bound the r.h.s.

A few computations show that $\frac{1-(2q+1)\alpha+q\alpha^2}{2(1-\alpha)} \leqslant 1 - (q+1)\alpha$. We thus get

$$
\delta \lesssim \frac{1}{D_F} \left( \lambda^{\xi_q(\alpha)L} + \lambda^{-\alpha L} \delta \right)
$$

which conclude the proof.

## A.7. Proof of Corollary 4.4

We have, using Lemma C.4 and $1_n^\top Y = 0$,

$$
\begin{aligned}
\left\| 1_n^\top B^{(H)} \right\|_2 &= \left\| 1_n^\top \left( F^{(L)} W^{(L)} - Y \right) \right\|_2 \\
&\leqslant n s^{L-1} D_\mathcal{X} \varepsilon_{W_L}
\end{aligned}
$$

And therefore, applying Theorem 4.3, the GNN is at a $\delta$-stationary point, with

$$
\delta \lesssim \lambda^{\xi_2(\alpha)L} + \lambda^{-2\alpha L} \varepsilon_{W_L}
$$

On the other hand, the loss is

$$
\mathcal{L}(H^{(L)}) = \frac{1}{2n} \left\| F^{(L)} W^{(L)} - Y \right\|_F^2 \qquad \geqslant \frac{1}{2n} \left( \|Y\|_F - \sqrt{n} s^{L-1} D_\mathcal{X} \varepsilon_{W_L} \right)^2 \geqslant \sigma_y^2/2 - \frac{1}{2} \sigma_y s^{L-1} D_\mathcal{X} \varepsilon_{W_L}
$$

using $\|Y\|_F = \sqrt{n} \sigma_y$ and $(a-b)^2 \geqslant a^2 - 2ab$ when $a, b \geqslant 0$.

### A.8. Proof of Proposition 4.5

Consider a regression problem such that the covariance of the node features is the identity $\frac{1}{n}X^\top X = Id_d$. Take a vector $w \in \mathbb{R}^d$ such that $\|w\| = 1$, and assume that the labels are $Y = X(w + w^\perp)$ where $w^\perp$ is a vector orthogonal to $w$ with $\|w^\perp\| = 1$.

Consider a linear MLP of depth $L$ with $\rho = Id$, with dimension $d_\ell = d$ for $\ell \leqslant k$, and $d_\ell = 1$ for $\ell > k$. Assume that all the weights are $W_\ell = Id$ (either in dimension $d$ or 1), except for the layer $k$ where the weights are $W_k = w$. We thus have $F^{(L)} = Xw$. Then, we have

$$\frac{\partial \mathcal{L}}{\partial W_L} = F^{(L)\top} B^{(L)} = \frac{1}{n} w^\top X^\top (Xw - Y) \tag{28}$$

$$= w^\top (w - w - w^\perp) = 0 \tag{29}$$

such that $W_L$ is at a stationary point. Moreover,

$$\frac{1}{n} \left\| F^{(L)} \right\|_F^2 = \frac{1}{n} \|Xw\|_F^2 = \|w\|^2 = 1 \tag{30}$$

so the output signal is indeed lower-bounded by $D_F = 1$.

However,

$$\left\| \frac{\partial \mathcal{L}}{\partial W^{(k)}} \right\| = \left\| (XW^{(0)} \dots W^{(k-1)})^\top B^{(L)} (W^{(k+1)} \dots W^{(L)})^\top \right\|$$

$$= \frac{1}{n} \left\| X^\top (Xw - Y) \right\| = \|w^\perp\| = 1$$

### A.9. Proof of Thm 4.3

We are going to prove the following, more detailed version of Thm 4.3.

**Theorem A.1.** *Assume that forward oversmoothing holds with rate $\lambda_f = \lambda$ and constant $C_f$. Under Assumptions 2.1 to 3.3, the GNN is at a global $\delta$-stationary point, with*

$$\delta \leqslant C_1(\lambda^{L-k} + \lambda^k) + C_2(s\lambda)^k + C_3\varepsilon_n \tag{31}$$

*where*

$$C_1 = \sqrt{n} D_\mathcal{X} \mu_\pi s^L A_L C_P$$

$$C_2 = \sqrt{n} D_\mathcal{X} \mu_\pi s^L A_L \frac{D_\rho C_f}{1 - \lambda s}$$

$$C_3 = n D_\mathcal{X} \mu_\pi s^L c_\pi$$

*If additionally $P$ is symmetric, the constants can be replaced with*

$$C_1 = D_\mathcal{X} s^L A_L$$

$$C_2 = D_\mathcal{X} s^L A_L \frac{D_\rho}{1 - \lambda s}$$

$$C_3 = D_\mathcal{X} s^L$$

*Proof of Thm A.1.* Our strategy is to use again the decomposition (25), but to bound directly the amplitude of $B^{(k)}$ instead of $\mathcal{E}_\pi(B^{(k)})$. This will involve the sum of the output error $\varepsilon_n := \left\| 1_n^\top B^{(L)} \right\|$.

**Last layers** In the last layers, the forward signal is almost constant, and therefore $F^\top B \approx v 1_n^\top B$ for some vector $v$, from which the sum $1_n^\top B$ appears. We thus directly examine the sum $1_n^\top B$.

For the first term in (25), since $P$ is a stochastic matrix,

$$\left\| 1_n^\top B_1 \right\| = \left\| 1_n^\top \left[ (P^\top)^{L-k} B^{(L)} V^{(L-1)} \dots V^{(k)} \right] \right\| = \left\| 1_n^\top \left[ B^{(L)} V^{(L-1)} \dots V^{(k)} \right] \right\| \leqslant \varepsilon_n s^{L-k}$$

Then, by Lemma C.2 and by (26): for any $q$,

$$\left\|1_n^\top((P^\top)^q M_k)\right\| = \left\|1_n^\top M_k\right\| \leqslant \sqrt{n}\left\|M_k\right\|_F \leqslant D_\rho C_f \mu_\pi A_L s^L \lambda_f^k$$

such that

$$\left\|1_n^\top B_2\right\| \leqslant D_\rho C_f \mu_\pi A_L s^L \sum_{\ell=k}^{L-1} \lambda^\ell s^{\ell-k} \leqslant \frac{D_\rho C_f \mu_\pi A_L s^L \lambda^k}{1-\lambda s}$$

From this we get, using $\left\|F^\top 1_n/n\right\| \leqslant \|F\|_{\infty,2}$ and $\|B\|_F / \sqrt{n} \leqslant \|B\|_{\infty,2}$, Lemma C.4 and C.3,

$$
\begin{aligned}
\left\|\frac{\partial \mathcal{L}}{\partial W^{(k)}}\right\|_F &= \left\|F^{(k)^\top} B^{(k)}\right\|_F \leqslant \left\|(JF^{(k)})^\top B^{(k)}\right\|_F + \left\|(J^\perp F^{(k)})^\top B^{(k)}\right\|_F \\
&= \left\|\frac{F^{(k)^\top} 1_n}{n} \cdot 1_n^\top B^{(k)}\right\|_F + \left\|B^{(k)}\right\|_F \sqrt{n}\mathcal{E}(F^{(k)}) \\
&\leqslant \left\|F^{(k)}\right\|_{\infty,2}\left\|1_n^\top B^{(k)}\right\| + n\left\|B^{(k)}\right\|_{\infty,2}\mathcal{E}(P\rho(H^{(k-1)})) \\
&\leqslant s^{k-1}D_\mathcal{X}\left(\varepsilon_n s^{L-k} + \frac{D_\rho C_f \mu_\pi A_L s^L \lambda^k}{1-\lambda s}\right) + \mu_\pi A_L s^{L-k} C_P \lambda C_f s^{k-1}\lambda_f^{k-1} \\
&\leqslant C_f \mu_\pi s^L A_L \left(\frac{D_\rho D_\mathcal{X} s^k}{1-\lambda s} + C_P\right)\lambda^k + s^L D_\mathcal{X}\varepsilon_n
\end{aligned}
$$

since $\mathcal{E}(PX) \leqslant C_P \lambda \mathcal{E}(X)$ similarly to (19).

Since $\lambda s < 1$, the rhs is decreasing with $k$ and therefore we obtain that: for all $k$ and all $\ell \geqslant k$,

$$\left\|\frac{\partial \mathcal{L}}{\partial H^{(\ell)}}\right\|_F \leqslant C_f \mu_\pi s^L A_L \left(\frac{D_\rho D_\mathcal{X}}{1-\lambda s}(\lambda s)^k + C_P \lambda^k\right) + s^L D_\mathcal{X}\varepsilon_n \tag{32}$$

**Middle and first layers**  In the middle layers, the forward signal is still oversmoothed, such that the norm of $M_k$ is still small. Moreover, the product of matrices $(P^\top)^{L-k}$ in the first term of (25) starts to become closer and closer to $\pi 1_n^\top$, such that $(P^\top)^{L-k}B^{(L)}$ *directly becomes related* to $\varepsilon_n = 1_n^\top B^{(L)}$. As a result, *the norm of $B^{(k)}$ itself* (and not of $1_n^\top B^{(k)}$ as in the last layers) becomes approximately related to $\varepsilon_n$ directly. We thus examine $\left\|B^{(k)}\right\|_F$.

For the first term of (25), by Assumption 2.2,

$$
\begin{aligned}
\|B_1\|_F &\leqslant \left(\left\|[(P^\top)^{L-k} - \pi 1_n^\top]B^{(L)}\right\|_F + \left\|\pi 1_n^\top B^{(L)}\right\|_F\right)s^{L-k} \\
&\leqslant \left(C_P \lambda^{L-k}\sqrt{n}\left\|B^{(L)}\right\|_{\infty,2} + c_\pi \varepsilon_n\right)s^{L-k} \\
&\leqslant \left(C_P \lambda^{L-k}\frac{A_L}{\sqrt{n}} + c_\pi \varepsilon_n\right)s^{L-k}
\end{aligned}
$$

Next, by Lemma C.3 and C.4, since $D_\pi^{-1}(P^\top)^q D_\pi$ is a stochastic matrix for any $q$, we directly examine

$$
\begin{aligned}
\left\|D_\pi^{-1}(P^\top)^q M_k\right\|_{\infty,2} &= \left\|D_\pi^{-1}(P^\top)^q D_\pi D_\pi^{-1}M_k\right\|_{\infty,2} \leqslant \left\|D_\pi^{-1}M_k\right\|_{\infty,2} \\
&= \left\|(J^\perp \rho_k) \odot (D_\pi^{-1}P^\top B^{(k+1)}W^{(k+1)^\top})\right\|_{\infty,2} \\
&\leqslant \left\|J^\perp \rho_k\right\|_\infty \left\|D_\pi^{-1}P^\top D_\pi D_\pi^{-1}B^{(k+1)}(W^{(k+1)})^\top\right\|_{\infty,2} \\
&\leqslant \left\|J^\perp \rho_k\right\|_F s\left\|D_\pi^{-1}B^{(k+1)}\right\|_{\infty,2} \leqslant \frac{A_L}{\sqrt{n}\pi_{\min}}s^{L-k}\mathcal{E}(\rho_k) \\
&\leqslant \frac{D_\rho A_L s^L C_f}{\sqrt{n}\pi_{\min}}\lambda^k
\end{aligned}
$$

by the same computation as in the proof of Thm 3.4 to bound $\left\|D_\pi^{-1} B^{(k+1)}\right\|_{\infty,2}$, and since $\left\|J^\perp \rho_k\right\|_F = \sqrt{n}\mathcal{E}(\rho_k)$.
Thus

$$\left\|D_\pi^{-1} B_2\right\|_{\infty,2} \leqslant \frac{D_\rho A_L s^L C_f}{\sqrt{n}\pi_{\min}} \sum_{\ell=k}^{L-1} \lambda^\ell s^{\ell-k} \leqslant \frac{D_\rho A_L s^L C_f}{\sqrt{n}\pi_{\min}(1-\lambda s)} \lambda^k \tag{33}$$

Then we directly relate the norm of $B^{(\ell)}$ to that of middle layers $B^{(k)}$ for $\ell < k$: by Lemma C.4,

$$\left\|B^{(\ell)}\right\|_F \leqslant \sqrt{n}\pi_{\max} s^{k-\ell} \left\|D_\pi^{-1} B^{(k)}\right\|_{\infty,2} \leqslant \sqrt{n}s^{k-\ell}(\mu_\pi \|B_1\|_F + \pi_{\max} \left\|D_\pi^{-1} B_2\right\|_{\infty,2}) \tag{34}$$

$$\leqslant \sqrt{n}s^{k-\ell} \left( \mu_\pi \left( C_P \lambda^{L-k} \frac{A_L}{\sqrt{n}} + c_\pi \varepsilon_n \right) s^{L-k} + \frac{D_\rho A_L s^L C_f \mu_\pi}{\sqrt{n}(1-\lambda s)} \lambda^k \right)$$

where $B_1, B_2$ are defined in (25), and using $\|\cdot\|_{\infty,2} \leqslant \|\cdot\|_F$. We then obtain that for all $k$ and all $\ell < k$,

$$\left\|F^{(\ell)^\top} B^{(\ell)}\right\|_F \leqslant \sqrt{n}s^\ell D_\mathcal{X} \left\|B^{(\ell)}\right\|_F$$

$$\leqslant \sqrt{n}D_\mathcal{X}\mu_\pi s^L \left( C_P \lambda^{L-k} A_L + \sqrt{n}c_\pi \varepsilon_n + \frac{D_\rho A_L C_f}{1-\lambda s}(s\lambda)^k \right) \tag{35}$$

Consider some index $k$. Combining (32) to bound the gradients with $\ell \geqslant k$ and (35) to bound the gradients with $\ell \leqslant k$, we obtain that for *all* indices $\ell$:

$$\left\|\frac{\partial \mathcal{L}}{\partial W^{(\ell)}}\right\| \leqslant C_1(\lambda^{L-k} + \lambda^k) + C_2(s\lambda)^k + C_3\varepsilon_n \tag{36}$$

where

$$C_1 = \sqrt{n}D_\mathcal{X}\mu_\pi s^L A_L C_P$$
$$C_2 = \sqrt{n}D_\mathcal{X}\mu_\pi s^L A_L \frac{D_\rho C_f}{1-\lambda s}$$
$$C_3 = nD_\mathcal{X}\mu_\pi s^L c_\pi$$

**Symmetric case** When $P$ is symmetric, we have $\|P\|_2 = 1$, $\mu_\pi = 1$, $c_\pi = 1/\sqrt{n}$, $C_P = C_f = 1$, and all assumptions are verified with $\lambda = \lambda_f$ the second-largest eigenvalue of $P$.

Hence, since $\|P\|_2 = 1$ we can replace (33) by:

$$\|B_2\|_F \leqslant \sum_{\ell=k}^{L-1} s^{\ell-k} \|M_\ell\| \leqslant \frac{D_\rho s^L A_L}{\sqrt{n}(1-\lambda s)} \lambda^k$$

by (26). Similarly, we can replace (34) by:

$$\left\|B^{(\ell)}\right\|_F \leqslant s^{k-\ell} \left\|B^{(k)}\right\|_F$$

$$\leqslant s^{k-\ell} \left( \left( \lambda^{L-k} \frac{A_L}{\sqrt{n}} + \varepsilon_n/\sqrt{n} \right) s^{L-k} + \frac{D_\rho s^L A_L}{\sqrt{n}(1-\lambda s)} \lambda^k \right)$$

and (35) is replaced by

$$\left\|F^{(\ell)^\top} B^{(\ell)}\right\|_F \leqslant \sqrt{n}s^\ell D_\mathcal{X} \left\|B^{(\ell)}\right\|_F$$

$$\leqslant D_\mathcal{X} s^L \left( \lambda^{L-k} A_L + \varepsilon_n + \frac{A_L}{1-\lambda s}(s\lambda)^k \right) \tag{37}$$

And therefore the bound (36) is valid with

$$C_1 = D_{\mathcal{X}} s^L A_L$$
$$C_2 = D_{\mathcal{X}} s^L A_L \frac{D_\rho}{1 - \lambda s}$$
$$C_3 = D_{\mathcal{X}} s^L$$

$\square$

We can now prove the variant in the paper Thm 4.3.

*Proof of Theorem 4.3.* Taking $s \leqslant \lambda^{-\alpha}$ and $k = \beta L$, the result of Thm A.1 can be rewritten as:

$$\left\| \frac{\partial \mathcal{L}}{\partial H^{(\ell)}} \right\|_F \lesssim \lambda^{((1-\alpha)\beta - q\alpha)L} + \lambda^{(1-\beta-q\alpha)L} + \lambda^{-\alpha L} \varepsilon_n$$

The terms that do not depend on $\varepsilon_n$ decrease with $L$ when $\frac{q\alpha}{1-\alpha} < \beta < 1 - q\alpha$. This is possible when $0 < 1 - (2q+1)\alpha + q\alpha^2$, which translates to having

$$\alpha < 1 + \frac{1}{2q} - \sqrt{1 + \frac{1}{4q^2}} \tag{38}$$

which is equal to $\approx 0.38$ for $q = 1$ and $\approx 0.21$ for $q = 2$. Choosing then $\beta = \frac{1}{2} \left( \frac{q\alpha}{1-\alpha} + 1 - q\alpha \right)$, we get the final rate

$$\left\| \frac{\partial \mathcal{L}}{\partial H^{(\ell)}} \right\|_F \lesssim \lambda^{\xi_q(\alpha)L} + \lambda^{-\alpha L} \varepsilon_n$$

$\square$

# B. Additional experiments

Here we illustrate backward oversmoothing on several datasets, with $P = D^{-1}A$. Recall that our goal is not to benchmark GNNs or oversmoothing mitigation strategies on many datasets, as this has been done many times in the literature (see (Rusch et al., 2023) and reference therein), but to briefly illustrate our theoretical results. The code is available at https://gitlab.inria.fr/nkeriven1/backward-oversmoothing-gnn.

**Settings.** We consider the classical Cora, Citeseer and Pubmed datasets (Yang et al., 2016), as well as the WikiCS dataset (Mernyei & Cangea, 2020) and two synthetic datasets of random graphs drawn from a Corrected Stochastic Block Model (CSBM) with two classes, for two different levels of density. We also test two heterophilic datasets, Chameleon and Squirrel (Rozemberczki et al., 2021) All graphs are transformed into undirected graphs and restricted to their largest connected component, such that $P = D^{-1}A$ is diagonalizable and Ass. 2.2 is satisfied with $\lambda_P = |\lambda_2| < 1$ the second-largest eigenvalue. Deep GNNs and MLP use 100 layers, except for the dense synthetic dataset where they use 40 layers (to avoid reaching machine-precision). We found that the phenomena described in this paper are exacerbated for the MSE loss (in particular the fact that the last layer $W_L$ quickly reaches a stationary point), hence for classification datasets we convert the labels into integers and cast only regression problems. As this paper is not concerned with generalization (and the considered models have all already been extensively benchmarked) but rather with gradient norms, we display the training loss and consider that every node is labelled for simplicity. To really assess oversmoothing and avoid vanishing signal/gradients as much as possible, we initialize the weights as $Id + \varepsilon \mathcal{N}$, where $\varepsilon$ is a small parameter and $\mathcal{N}$ contains independent normal variables. This way, $s \approx 1$, and our theorems are satisfied while avoiding vanishing gradients. We use a Leaky ReLU non-linearity $\rho$.

**Forward oversmoothing and rates of convergence.** In table 1, we compute $\lambda_2$ and $\left\| J^\perp P J^\perp \right\|_2$ for all datasets. We first observe that (11) is satisfied only for the *dense* synthetic dataset. However, even if the conditions of Thm. 3.2 are not satisfied, we observe in Fig. 2 (left) that forward oversmoothing indeed seems to hold. With a simple regression, we then estimate the rate of convergence $\lambda_f$, and we observe that $\lambda_f \approx \lambda_P$. While it seems that the higher the density, the lower $\lambda_P$,

| Dataset | Density | $\lambda_f$ (est.) | $\lambda_P = |\lambda_2|$ | $\left\|J^\perp P J^\perp\right\|_2$ |
|---|---|---|---|---|
| Synthetic (sparse) | $8.80e-3$ | 0.906 | 0.923 | 1.72 |
| Synthetic (dense) | $2.99e-2$ | 0.744 | 0.742 | 0.788 |
| Cora | $1.64e-3$ | 0.953 | 0.971 | 4.48 |
| Citeseer | $1.64e-3$ | 0.960 | 0.984 | 4.43 |
| Pubmed | $2.28e-3$ | 0.952 | 0.986 | 6.98 |
| WikiCS | $3.37e-3$ | 0.901 | 0.917 | 9.55 |
| Chameleon | $1.21e-2$ | 0.958 | 0.895 | 6.83 |
| Squirrel | $1.47e-2$ | 0.927 | 0.956 | 4.14 |

*Table 1.* Properties of the different datasets. The density is the number of edges over the number of possible edges $n(n-1)/2$. The rate $\lambda_f$ is estimated by a regression on the forward oversmoothing curve (Fig. 2, left).

which is intuitively not surprising since it is related to the spectral gap of $Id - P$, the value of $\left\|J^\perp P J^\perp\right\|_2$ does not seem to be directly related to the density of the graph (except for the synthetic dense graph), which hints at a more complex situation. Overall, as observed in the paper, the general case "non-linear $\rho$, non-symmetric $P$" remains an important and non-trivial missing piece of forward oversmoothing.

**GNN training, comparison with MLP.** In Fig. 2 and 4, we train a deep GNN and a deep MLP on our datasets. We first observe (left) that backward oversmoothing indeed holds *in the middle layers* as shown by Thm. 3.4. Then, training our models and monitoring both the loss and the norm of the gradients, it indeed seems that deep GNNs tend to quickly reach a stationary point with a high loss, compared to their MLP counterpart. As shown by the theory, this effect is greatly accentuated when $\lambda$ is smaller (Tab. 1), in particular in the synthetic datasets and WikiCS dataset. On the three other datasets, this is less clear, as $\lambda$ is quite close to 1.

**Skip connections.** In Fig. 3, we compare GNN with and without skip connections. We first observe (left) that skip connections indeed allow to largely counter both forward and backward oversmoothing. However, stability of the overall model largely depends on $s$: a too large $s$ can lead to explosion of both forward and backward signal. With a balanced $s$ around $s = \frac{5}{L}$, skip connections also appear to avoid the stationary points of regular GNNs (center and right). However, when using skip connections naively like this (without normalization, etc.), the loss still tends to plateau compared to the MLPs of Fig. 2. Future work will have to theoretically analyze (similar to (Chen et al., 2025)), and hopefully eventually improve, the use of skip connections from an optimization point of view.

**Effect of learning rate.** In Fig. 5, we test three different learning rates and examine the gradient norms and loss for deep GNNs. While the learning rate largely affects the beginning of the training, it seems that most of the time the GNN will again fall into a spurious stationary point with high loss but gradients. Exceptions include when the learning rate is too low and the weights barely moves, or too high and the training becomes very noisy. Both cases also incur high losses.

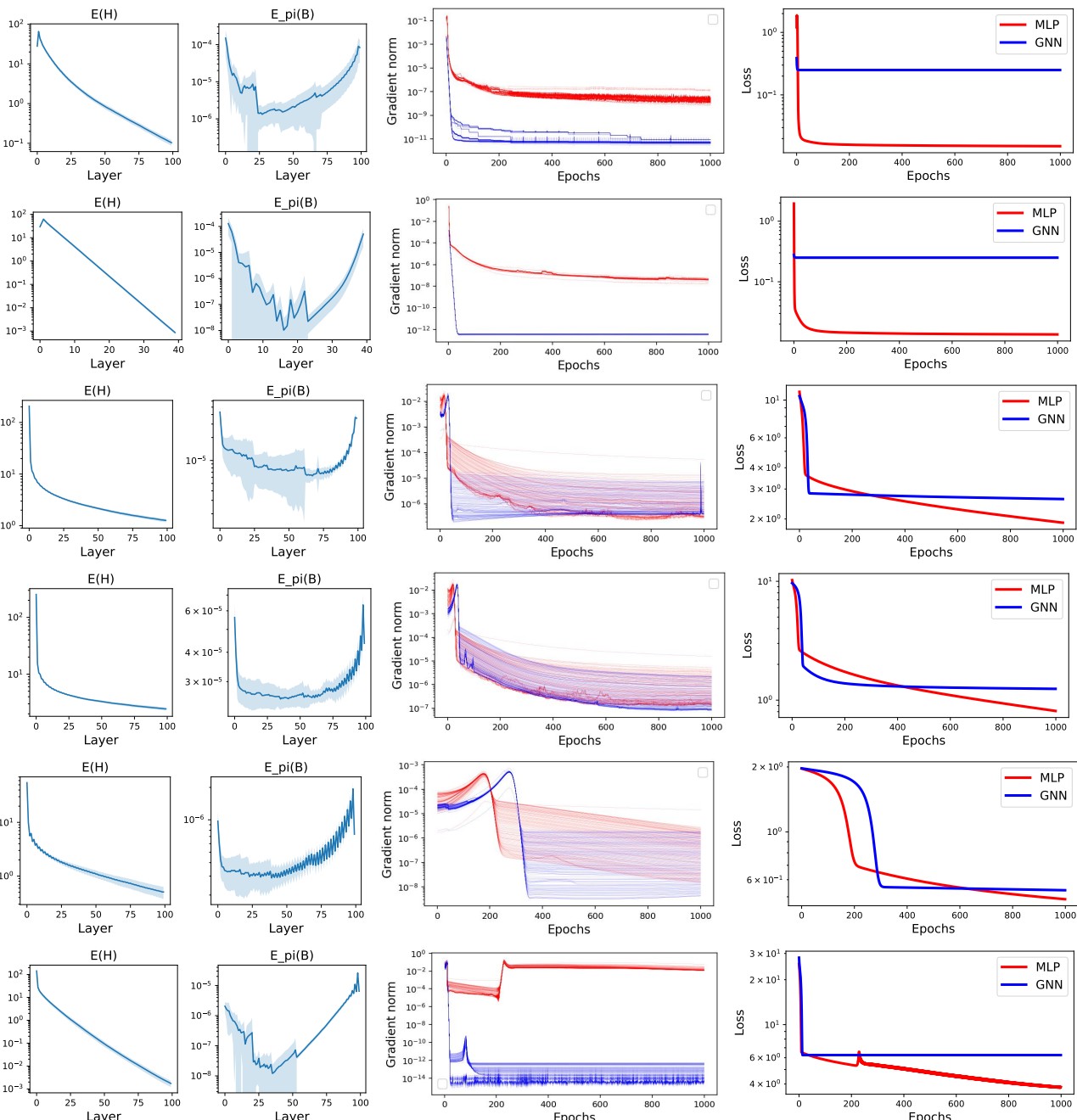

*Figure 2.* **From top to bottom**: datasets: Synthetic (sparse), Synthetic (dense), Cora, Citeseer, Pubmed, WikiCS. **Left**: illustration of forward oversmoothing by measuring $\mathcal{E}(H^{(k)})$ (left) and backward oversmoothing by measuring $\mathcal{E}_\pi(B^{(k)})$ (right). Averaged over 10 random initializations. **Center**: norms of gradients $\partial\mathcal{L}/\partial W^{(k)}$ for each layer, for an MLP (red) and GNN (blue) with 100 layers (except for the Synthetic Dense dataset for which we use 40 layers). **Right**: corresponding losses for MLP and GNN.

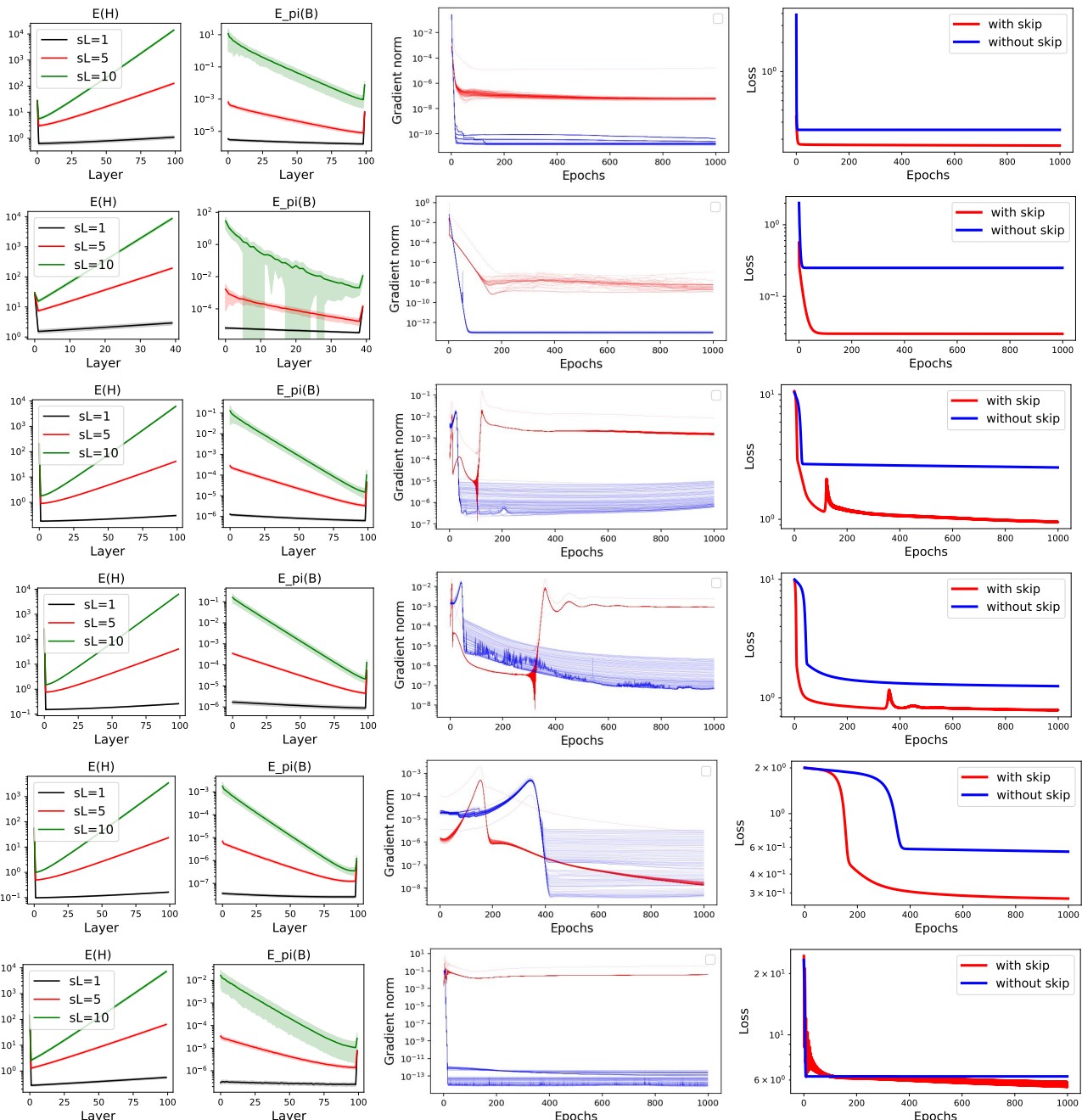

*Figure 3.* Illustration of skip connections. **From top to bottom**: datasets: Synthetic (sparse), Synthetic (dense), Cora, Citeseer, Pubmed, WikiCS. **Left**: for GNNs with skip connections, illustration of forward oversmoothing by measuring $\mathcal{E}(H^{(k)})$ (left) and backward oversmoothing by measuring $\mathcal{E}_\pi(B^{(k)})$ (right). Averaged over 10 random initializations, for three levels of weight amplitude: $s = \frac{1}{L}$, $s = \frac{5}{L}$, $s = \frac{10}{L}$. **Center**: norms of gradients $\partial\mathcal{L}/\partial W^{(k)}$ for each layer, for a GNN with skip connections and $s = \frac{5}{L}$ at initialization (red), and GNN without skip connections (blue) with 100 layers (except for the Synthetic Dense dataset for which we use 40 layers). **Right**: corresponding losses for MLP and GNN.

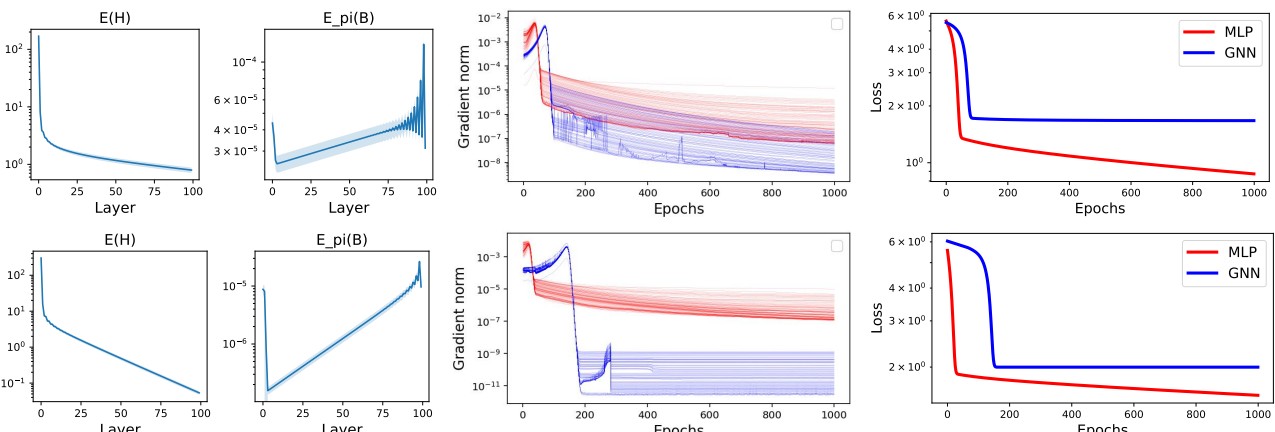

*Figure 4.* Similar to Figure 2 for the Chameleon (top) and Squirrel (bottom) datasets.

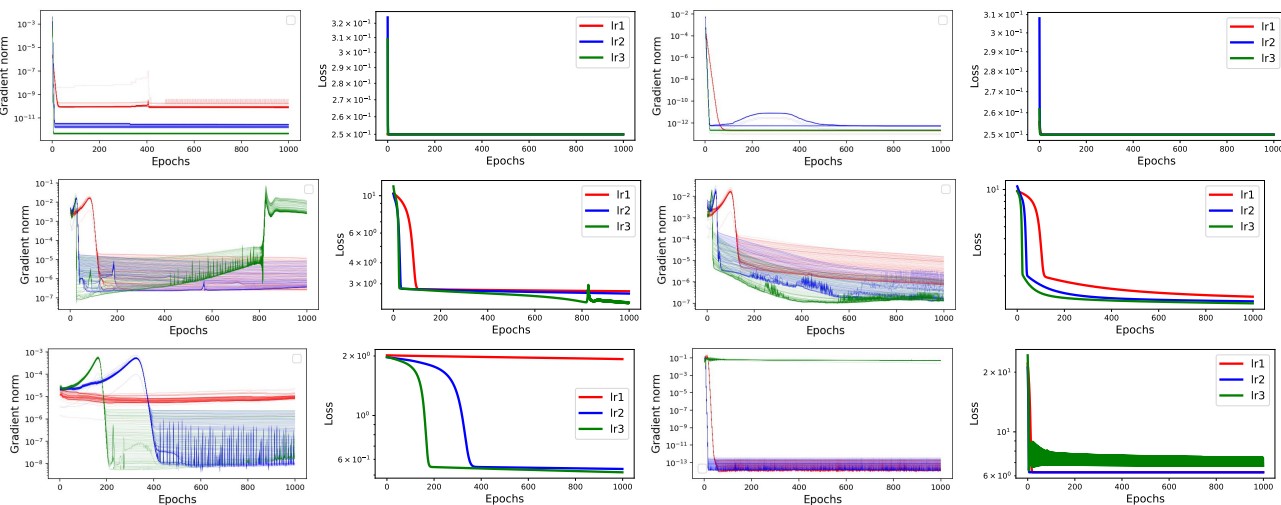

*Figure 5.* **From top to bottom, left to right** (two-by-two): datasets: Synthetic (sparse), Synthetic (dense), Cora, Citeseer, Pubmed, WikiCS. Each time, norms of gradients $\partial \mathcal{L} / \partial W^{(k)}$ for each layer (left) and corresponding losses (right) for deep GNNs, for three different learning rates $lr_1 = 10^{-4}$, $lr_2 = 3 \cdot 10^{-4}$ (choice in the other figures), $lr_3 = 6 \cdot 10^{-4}$.

# C. Technical Lemma

**Lemma C.1.** *Given $u \in \mathbb{R}^n$ and $X \in \mathbb{R}^{n \times d}$, decompose orthogonally each column of $X$, that is $X = uv^\top + X^\perp$, where $v \in \mathbb{R}^d$ and $u^\top X^\perp = 0$. Then*

$$\mathcal{E}_u(X) = n^{-1/2} \left\| X^\perp \right\|_F \tag{39}$$

*In particular:*

1. $\mathcal{E}_u(X + Y) \leqslant \mathcal{E}_u(X) + \mathcal{E}_u(Y)$

2. $\mathcal{E}_u(X) \leqslant n^{-1/2} \left\| X \right\|_F$

3. for all $v$, $\mathcal{E}_u(X) = \mathcal{E}_u(X + uv^\top)$

4. $\mathcal{E}_u(XW) = n^{-1/2} \left\| X^\perp W \right\|_F \leqslant \mathcal{E}_u(X) \left\| W \right\|_2$

*Proof.* The proof is obvious as $\mathcal{E}_u(x) = n^{-1/2} \min_c \| x - cu \|$, computes the orthogonal projection of $x$ on $span(u)^\perp$. All subsequent properties are immediate, using $\| XW \|_F \leqslant \| X \|_F \| W \|_2$ for the last one. $\qquad \square$

**Lemma C.2.** *For two matrices $A, B \in \mathbb{R}^{n \times d}$ we have*

$$\left\| (A \odot B)^\top 1_n \right\|_2 \leqslant \sqrt{n} \left\| B \right\|_\infty \left\| A \right\|_F \tag{40}$$

*Moreover, denoting $J = 1_n 1_n^\top / n$,*

$$\left\| ((JA) \odot B)^\top 1_n \right\|_2 \leqslant \left\| A \right\|_\infty \left\| B^\top 1_n \right\|_2 \tag{41}$$

*Proof.* Simply

$$\begin{aligned}
\left\| (A \odot B)^\top 1_n \right\|_2^2 &= \sum_{j=1}^d \left( \sum_{i=1}^n a_{ij} b_{ij} \right)^2 \\
&\leqslant \sum_{j=1}^d \left( \sum_{i=1}^n a_{ij}^2 \right) \left( \sum_{i=1}^n b_{ij}^2 \right) \leqslant n \left\| B \right\|_\infty^2 \left\| A \right\|_F^2
\end{aligned}$$

and

$$\begin{aligned}
\left\| ((JA) \odot B)^\top 1_n \right\|_2^2 &= \sum_{j=1}^d \left( \sum_{i=1}^n \left( \frac{1}{n} \sum_k a_{kj} \right) b_{ij} \right)^2 \\
&= \sum_{j=1}^d \left( \frac{1}{n} \sum_k a_{kj} \right)^2 \left( \sum_{i=1}^n b_{ij} \right)^2 \\
&\leqslant \left\| A \right\|_\infty^2 \left\| B^\top 1_n \right\|_2^2
\end{aligned}$$

$$\square$$

**Lemma C.3.** *For all $k$ we have*

$$\left\| P^k X \right\|_{\infty,2} \leqslant \left\| X \right\|_{\infty,2}, \qquad \left\| (P^\top)^k X \right\|_{\infty,2} \leqslant \mu_\pi \left\| X \right\|_{\infty,2} \tag{42}$$

*We also have*

$$\left\| XW \right\|_{\infty,2} \leqslant \left\| W \right\|_2 \left\| X \right\|_{\infty,2} \tag{43}$$

*Proof.* It is immediate that to a stochastic matrix $P$,

$$\|PX\|_{\infty,2} = \max_i \left\| \sum_j P_{ij} X_{j,:} \right\|_2$$

$$\leqslant \|X\|_{\infty,2} \max_i \sum_j |P_{ij}| = \|X\|_{\infty,2}$$

and $P^k$ is stochastic.

Then, since $D_\pi^{-1}(P^\top)^k D_\pi$ is stochastic,

$$\left\|(P^\top)^k X\right\|_{\infty,2} \leqslant \pi_{\max} \left\|D_\pi^{-1}(P^\top)^k D_\pi D_\pi^{-1} X\right\|_{\infty,2} \leqslant \pi_{\max} \left\|D_\pi^{-1} X\right\|_{\infty,2} \leqslant \mu_\pi \|X\|_{\infty,2}$$

The second inequality is immediate by

$$\|XW\|_{\infty,2} = \max_i \left\|W X_{i,:}^\top\right\|_2 \leqslant \|W\|_2 \|X\|_{\infty,2}$$

$\square$

**Lemma C.4.** *For all $k$ we have:*

$$\left\|F^{(k+1)}\right\|_{\infty,2} \leqslant \left\|X^{(k)}\right\|_{\infty,2} \leqslant \left\|H^{(k)}\right\|_{\infty,2} \leqslant s^k D_\mathcal{X} \tag{44}$$

*and*

$$\left\|B^{(L)}\right\|_{\infty,2} \leqslant \frac{A_L}{n}, \qquad \left\|B^{(k)}\right\|_{\infty,2} \leqslant \frac{\mu_\pi A_L}{n} s^{L-k}$$

*where $A_L = s^L D_\mathcal{X} D_\mathcal{L} + D'_\mathcal{L}$.*

*Finally, for all $\ell > k$:*

$$\left\|D_\pi^{-1} B^{(k)}\right\|_{\infty,2} \leqslant \left\|D_\pi^{-1} B^{(\ell)}\right\|_{\infty,2} s^{\ell-k} \tag{45}$$

*Proof.* By recursion, since $|\rho(x)| \leqslant |x|$ and by Lemma C.3,

$$\left\|H^{(k)}\right\|_{\infty,2} = \left\|P\rho(H^{(k-1)})W^{(k)}\right\|_{\infty,2} \leqslant s \left\|H^{(k-1)}\right\|_{\infty,2} \leqslant s^k \left\|X^{(0)}\right\|_{\infty,2} \leqslant s^k D_\mathcal{X}$$

and easily

$$\left\|F^{(k+1)}\right\|_{\infty,2} = \left\|PX^{(k)}\right\|_{\infty,2} \leqslant \left\|X^{(k)}\right\|_{\infty,2} = \left\|\rho(H^{(k)})\right\|_{\infty,2} \leqslant \left\|H^{(k)}\right\|_{\infty,2}$$

In the same fashion, since $\rho' \leqslant 1$ and using Lemma C.3, since for a diagonal matrix $D$ we have $D(A \odot B) = A \odot (DB)$, and $D_\pi^{-1} P^\top D_\pi$ is stochastic,

$$\left\|D_\pi^{-1} B^{(k)}\right\|_{\infty,2} = \left\|\rho'(H^{(k)}) \odot \left(D_\pi^{-1} P^\top B^{(k+1)}(W^{(k+1)})^\top\right)\right\|_{\infty,2} \tag{46}$$

$$\leqslant \left\|D_\pi^{-1} P^\top D_\pi D_\pi^{-1} B^{(k+1)}(W^{(k+1)})^\top\right\|_{\infty,2} \tag{47}$$

$$\leqslant s_{k+1} \left\|D_\pi^{-1} B^{(k+1)}\right\|_{\infty,2} \leqslant s^{(k+1:\ell+1)} \left\|D_\pi^{-1} B^{(\ell)}\right\|_{\infty,2} \tag{48}$$

choosing $\ell = L$ and $\left\|D_\pi^{-1} B^{(L)}\right\|_{\infty,2} \leqslant \frac{1}{\pi_{\min}} \left\|B^{(L)}\right\|_{\infty,2}$ and $\left\|B^{(L)}\right\|_{\infty,2} \leqslant \frac{1}{n}(D_\mathcal{L} \left\|H^{(L)}\right\|_{\infty,2} + D'_\mathcal{L}) \leqslant \frac{1}{n}(s^L D_\mathcal{X} D_\mathcal{L} + D'_\mathcal{L})$ we obtain the final bound.

$\square$

