# OpenReview forum: "Backward Oversmoothing: why is it hard to train deep Graph Neural Networks?"
_ICML.cc/2026/Conference — ICML 2026 regular_

### Official Review · Reviewer_FnwR · 2026-03-12

**Soundness:** 3
**Presentation:** 3
**Significance:** 3
**Originality:** 3
**Overall Recommendation:** 4
**Confidence:** 3

**Summary:**

This paper investigates the oversmoothing problem in Graph Neural Networks (GNNs) from an optimization perspective. During forward propagation, node representations tend to converge as the number of layers increases ; similarly, during backpropagation, error signals also become smoothed due to the transpose operation of the graph propagation matrix, a phenomenon that is particularly prominent in the middle layers. Theoretical analysis demonstrates that, under the influence of non-linear activation functions, backward oversmoothing causes the gradients across layers to become uniform, making the optimization process highly susceptible to falling into spurious stationary points. Specifically, Theorem 4.2 indicates that once the final layer of the GNN converges, the gradients in all layers will vanish simultaneously, thereby hindering further optimization. Furthermore, comparative experiments confirm that this phenomenon is specific to GNNs and does not occur in standard Multi-Layer Perceptrons (MLPs).

**Compliance With Llm Reviewing Policy:**

Affirmed.

**Final Justification:**

It seems the current manuscript could be further refined in experiments, and I maintain my original score.

**Key Questions For Authors:**

1.Does this theory also apply to more complex graph neural network architectures?

2.Can existing methods for alleviating oversmoothing also effectively mitigate backward oversmoothing?

3.Although Proposition 3.6 theoretically demonstrates that backward oversmoothing does not necessarily lead to the vanishing gradient problem, in the experiments shown in Figure 1, the gradient norm of GNNs still differs significantly from that of MLPs and remains close to zero. When the model becomes trapped in spurious fixed points, what is the actual frequency of gradients not completely vanishing?

**Limitations:**

yes

**Strengths And Weaknesses:**

Strengths:

1. This paper presents a pioneering and systematic investigation into the 'backward oversmoothing' phenomenon, successfully extending the research paradigm of oversmoothing from the forward pass to the backward optimization process. This novel perspective is highly significant. Beyond establishing a rigorous definition of backward oversmoothing, the authors provide an in-depth analysis of the attenuation and convergence patterns of gradient flow across successive layers. By characterizing the properties of spurious stationary points, the work elucidates the underlying dynamical mechanisms that lead to optimization bottlenecks in deep GNNs. The theoretical framework is comprehensive, supported by meticulous mathematical derivations.
2. Comparative analysis confirms that this phenomenon is specific to GNNs and does not occur in MLPs, highlighting its deep connection to the graph propagation mechanism. Such a discovery provides significant implications for advancing the design of robust optimization algorithms for deep GNNs.

Weaknesses:

1. The majority of experimental results and graphical details are relegated to the appendix, while the empirical analysis within the main text remains relatively sparse. This, to some extent, hinders the reader's intuitive understanding of how the proposed theoretical framework manifests in real-world data performance.
2. The paper primarily investigates and formalizes the phenomenon of 'backward oversmoothing' in GNNs from a theoretical standpoint, yet it falls short of proposing concrete algorithmic solutions. As a study centered on mechanistic explanation, its impact is somewhat constrained by the absence of targeted mitigation strategies. Furthermore, the manuscript does not discuss whether established techniques for alleviating forward oversmoothing offer any synergistic benefits for the backward pass—a critical question that warrants further investigation.

---

> ### Author Rebuttal · Authors · 2026-03-30
>
> W1. Indeed, due to space constraints, we had to relegate the majority of the numerics to the appendix. We still believe that the intuitions behind the theorems, and more importantly behind the sketch of proof, is the focus of the paper here, however if some space can be gained after revising the paper, some of the numerics may be integrated to the main paper.
>
> W2. Indeed, we only brushed potential mitigating strategies, such as skip connections. As the presented analysis for the vanilla case was already quite involved, both in terms of technical content and interesting intuitions, we elected to leave it for future work. As we discussed in the paper, existing theoretical analyses for skip connections and vanilla *forward* oversmoothing are long and complex [1], and their adaptation to the backward case is a major outlook that will be the topic of its own paper.
>
> Q1. This is an interesting point that was also raised by reviewer uYfD. For us, the next natural step would be attention-based GNNs, since they are substantially different from the fixed message-passing matrix $P$ analyzed here, and used in practice. As shown by [2], attention-based GNNs may still be subjected to forward oversmoothing in the linear case $\rho=Id$. However their gradients have very different expressions, since the attention coefficients are also trained. Whether backward oversmoothing holds in this case is an open question, whose study could well explain why attention models deal with this problem more efficiently. We will add this outlook in the discussion.
>
> Q2. Fig 3 seems to show that skip connections alleviate, to some extent, the backward oversmoothing problem, but require some care in the absence of more complex normalization. Its analysis is left for future work (see above).
>
> Q3. Prop 3.6 shows indeed that backward oversmoothing is not *automatically* vanishing gradients (e.g. at initialization), but the rest of the paper shows that, however, as soon as the last layer is trained (an easy-to-reach condition in practice), gradients *do* indeed vanish. This is what is observed in practice: gradients for deep GNNs vanish almost immediately after a few steps of training, and the GNN is trapped is a "bad" stationary point with small gradients. We will clarify this.
>
>
> [1] Chen et al., Residual connections provably mitigate oversmoothing in graph neural networks. 2025
>
> [2] Wu et al., Demystifying oversmoothing in attention-based graph neural networks. NeurIPS 2023.

---

> > ### Author Rebuttal · Reviewer_FnwR · 2026-04-02
> >
> > I appreciate your response and clarification. Having read the responses (including those to other reviewers), it seems the current manuscript could be further refined in experiments.

---

> > > ### Author Response · Authors · 2026-04-03
> > >
> > > Thank you for your feedback. As mentioned in the reply to reviewer uYfD, we will indeed do our best to complete the numerical experiments in the Appendix, focusing on illustrating our theory (since the oversmoothing phenomenon itself has already been extensively benchmarked). In particular, in addition to more datasets with different spectral gaps, we will add illustrations regarding the optimization algorithm itself and its hyperparameters. Preliminary experiments (we cannot include figures here) seem to show that
> > > - changing the learning rate can delay falling into the ``spurious'' stationary points, but it consistently happens after a while.
> > > - Optimization algorithms other than vanilla SGD are more or less always subjected to the same phenomenon, but we however observe that momentum-based approaches such as Adam (unsurprisingly) add fluctuations in the gradient norm that may delay running into the stationary point, nevertheless it does not improve the loss. Naturally, such algorithms with momentum are still out-of-reach of our theory for now.
> > >
> > > We will complete the experiments in this direction in the final version. We welcome any suggestions for other experiments to run.

---

### Official Review · Reviewer_J7FX · 2026-03-12

**Soundness:** 3
**Presentation:** 3
**Significance:** 3
**Originality:** 4
**Overall Recommendation:** 5
**Confidence:** 4

**Summary:**

This paper studies the oversmoothing phenomenon in deep graph neural networks (GNNs) from an optimization perspective and introduces the concept of backward oversmoothing, where the backpropagated error signal becomes increasingly smooth across nodes as it propagates from the output layer to earlier layers. Under the assumptions of forward oversmoothing and smooth activations, the paper derives an upper bound on backward oversmoothing and analyzes its implications for the optimization dynamics of deep GNNs. In particular, the authors show that in sufficiently deep vanilla GNNs, once the last layer becomes near-stationary, the entire network can easily enter a near-stationary region due to the smoothing of backward signals. The paper also provides an existence result showing that such near-stationary regions may occur at high loss in regression settings, and presents a proposition suggesting that a similar mechanism does not generally arise in MLPs.
Overall, the paper proposes an interesting optimization perspective for understanding the training difficulties of deep GNNs.

**Compliance With Llm Reviewing Policy:**

Affirmed.

**Key Questions For Authors:**

Q1. The main theoretical results rely on the assumption that forward oversmoothing already occurs. Could the authors provide empirical evidence showing that this assumption typically holds in practical GNN training?

Q2. For Theorem 4.2, could the authors provide more direct experimental validation, e.g., by monitoring layer-wise gradient norms during training?

Q3. Assumption 3.3 requires the activation derivative to be Lipschitz, which excludes non-smooth activations such as ReLU. However, the experiments appear to use Leaky ReLU. Could the authors comment on whether the theoretical analysis extends to such piecewise linear activations?

**Limitations:**

yes

**Strengths And Weaknesses:**

**Strengths**

- Training deep GNNs has long been challenging in graph learning. While most existing explanations focus on forward oversmoothing, this work introduces backward oversmoothing from the perspective of backpropagated signals, offering a new angle for understanding GNN optimization behavior.
- By analyzing the recursive structure of backpropagation, the paper derives an upper bound on intermediate-layer gradients. Theorem 3.4 presents a decomposition where one term decays with k and the other with L−k, providing intuition for why backward oversmoothing is most pronounced in middle layers.
- Figure 1 qualitatively illustrates the evolution of forward representations and backward signals with depth and contrasts them with MLP behavior, helping convey the main intuition of the paper.

**Weaknesses**

- The theoretical results rely on the assumption that forward oversmoothing already occurs. However, forward oversmoothing itself is not theoretically established in the general nonlinear non-symmetric setting considered in the paper. It would be helpful if the authors could clarify how essential this assumption is, and whether the proposed mechanism could arise even when forward oversmoothing is weak.
- The experiments mainly provide qualitative demonstrations of the phenomenon. Direct validation of the core theoretical claims (e.g., Theorem 4.2) is limited. Additional experiments more closely aligned with the theoretical results would strengthen the empirical support.
- Recent work [1] argues that trainability issues rather than oversmoothing are the dominant reason for performance degradation in deep GNNs. It would be helpful for the authors to clarify how the proposed backward oversmoothing mechanism relates to this alternative explanation.

References:

[1]Peng J, Lei R, Wei Z. Beyond over-smoothing: Uncovering the trainability challenges in deep graph neural networks[C]//Proceedings of the 33rd ACM International Conference on Information and Knowledge Management. 2024: 1878-1887.

---

> ### Author Rebuttal · Authors · 2026-03-30
>
> We thank the reviewer for their questions and remarks. We comment and answer below.
>
> 1. (and W1) Indeed our analysis strongly relies on forward overmoothing, and does not hold without it. However, as we observe in Figure 2 (left), forward oversmoothing seem to always hold even in the general case, with very small variance across different initializations. Regarding the relative "strength" of forward vs backward oversmoothing, our theorems show that the rates $\lambda$ are related, even if empirical observations (Fig 2 again) is naturally slightly more noisy.
>
> 2. (and W2) Layer-wise gradient monitoring is provided in Fig 2, center. For many datasets, depending on the rate of oversmoothing, we indeed see the GNN quickly reach a bad stationary point at all layers.
>
> 3. Indeed, our analysis would technically exclude ReLU-like non-linearities. We are not sure if and how extending our theory to piecewise-linear $\rho$ would be feasible, but this is an interesting outlook, as there are several works on this for regular NNs. Nevertheless, we still choose to use ReLU-like $\rho$ in the experiments since it is far more used in practice. We choose LeakyReLU instead of ReLU to avoid zeroing out too much of the signal, which would have somewhat interfered with the monitoring of the stationary points.
>
> W3. Thank you for providing this very relevant reference, that we will add. It seems that the authors argue that training many weights ("deep MLP") is also a challenge, which is somewhat a complementary point of view to ours: on the "right" multiplicative side, training many weights is naturally a challenge that also arise for MLPs, but on the "left" multiplicative side, backward oversmoothing may lead to unavoidable bad stationary points that are, this time around, specific to GNNs. The overall conclusion is however somewhat in line with ours, that *optimization* is a crucial point (if not *the* crucial point) in deep GNNs. We will examine this reference more in details and add a discussion.

---

> > ### Author Rebuttal · Reviewer_J7FX · 2026-04-03
> >
> > All my concerns are well addressed.

---

### Official Review · Reviewer_uYfD · 2026-03-13

**Soundness:** 3
**Presentation:** 2
**Significance:** 3
**Originality:** 3
**Overall Recommendation:** 4
**Confidence:** 4

**Summary:**

This paper claims that backward oversmoothing, arises from the interaction of forward smoothing with nonlinear activations, and that it has important optimization consequences. The authors also claim that this phenomenon is specific to GNNs and does not arise in MLP based systems.

**Compliance With Llm Reviewing Policy:**

Affirmed.

**Final Justification:**

Assuming the additional experiments the authors reported, I adjust my score

**Key Questions For Authors:**

1.	How sensitive is backward oversmoothing to the choice of activation function ρ?
2.	Does the severity of backward oversmoothing depend primarily on graph topology (e.g., spectral gap of P)?
3.	Could your framework explain why some GNN variants (e.g., attention based GNNs) behave differently when scaled deeply?
4.	Is there empirical evidence showing gradients collapsing even when forward features remain distinguishable?
5.     The experimental part must be extended

**Limitations:**

The experimental part is weak.

**Strengths And Weaknesses:**

The paper presents a novel idea and the idea provides an important contribution. The paper provides a strong theoretical results such as, the characterization of the smoothed limit, and in particular the backward signal. The paper also discussed the  interaction between forward and backward smoothing creates gradient collapse in intermediate layers.I also think that the proof that this phenomenon is not present in MLPs has a great value.

On the down side. I would expect the paper to extend the discussion on the practical implications of the finding.
I believe that the experimental section must be extended before the paper could be presented.

---

> ### Author Rebuttal · Authors · 2026-03-30
>
> We thank the reviewer for their positive view on our contribution, and answer their question below.
>
> Regarding the reviewer's main concern (Q5), the numerical experiments indeed have a rather illustrative role here, since the main focus is theoretical and space constraints relegate almost all numerics to the appendix. In particular, since this paper does not propose a *new* methodology but analyse well-established existing practices, we felt that it did not required extensive benchmarking but rather relevant illustrations of the theorems. Nevertheless, following the reviewer's remark 2., we will extend the existing experiments with more diverse datasets, including heterophilic ones, to better illustrate the role of the spectral gap, and of the various quantities in Table 1.
>
> Other questions:
> 1. In theory, as long as $\rho$ satisfies assumption 2.3 and 3.3, its choice only affect various multiplicative constants. In practice, the sensitivity if even lesser: all the phenomena are still observed for ReLU or LeakyReLU (which are technically not even differentiable), and since the forward signal is almost constant for deep layers, the term $\rho'(F) \odot$ that appears in the backward equations plays little role.
>
> 2. Indeed, when $\rho=Id$ or when $P$ is symmetric, the smoothing rate $\lambda_f = \lambda_P = \lambda_2$ is exactly the spectral gap of $P$. In the general case, $\lambda_f$ can only be estimated, but as seen in Table 1 it is strongly linked to the spectral gap. We will clarify this.
>
> 3. This is a good point, thank you for raising this. As shown by [1], attention-based GNNs may still be subjected to forward oversmoothing in the linear case $\rho=Id$. However their gradients have substantially different expressions, since the attention coefficients are also trained.  Whether backward oversmoothing holds in this case is an open question, whose study could well explain why attention models deal with this problem more efficiently. We will add this important outlook in the discussion.
>
> 4. Not to our knowledge, unless we are talking about "regular" vanishing gradients due to the usual reasons (random weights with unappropriate variance, sinusoidal non-linearities...), that would also happen for MLPs. The "spurious" stationary points for GNNs that we identified here are strongly linked to both forward and backward oversmoothing.
>
> 5. See above.
>
>
> [1] Wu et al., Demystifying oversmoothing in attention-based graph neural networks. NeurIPS 2023.

---

> > ### Author Rebuttal · Reviewer_uYfD · 2026-03-31
> >
> > I still think that the experiment part need to be extended

---

> > > ### Author Response · Authors · 2026-04-03
> > >
> > > Thank you for your feedback. We will indeed do our best to complete the numerical experiments in the Appendix, focusing on illustrating our theory (since the oversmoothing phenomenon itself has already been extensively benchmarked). In particular, in addition to more datasets with different spectral gaps, we will add illustrations regarding the optimization algorithm itself and its hyperparameters. Preliminary experiments (we cannot include figures here) seem to show that
> > > - changing the learning rate can delay falling into the ``spurious'' stationary points, but it consistently happens after a while.
> > > - Optimization algorithms other than vanilla SGD are more or less always subjected to the same phenomenon, but we however observe that momentum-based approaches such as Adam (unsurprisingly) add fluctuations in the gradient norm that may delay running into the stationary point, nevertheless it does not improve the loss. Naturally, such algorithms with momentum are still out-of-reach of our theory for now.
> > >
> > > We will complete the experiments in this direction in the final version. We welcome any suggestions for other experiments to run.

---

### Official Review · Reviewer_nRS2 · 2026-03-18

**Soundness:** 2
**Presentation:** 3
**Significance:** 3
**Originality:** 4
**Overall Recommendation:** 5
**Confidence:** 4

**Summary:**

The paper studies the phenomenon of "backward oversmoothing", a term the authors coined to describe the difficulties encountered empirically when training deep graph neural networks during backward passes. The authors provide elegant theory that attempts to explain this phenomenon, and identify certain key elements and observations which are specific to GNNs and not to feedforward NNs.

**Compliance With Llm Reviewing Policy:**

Affirmed.

**Final Justification:**

updated in light of rebuttal.

**Key Questions For Authors:**

1. I believe the relationships between forward oversmoothing and backwards oversmoothing could be more explicitly presented/clarified.
2. I am not entirely convinced by the choice of wording "spurious stationary point"...what is so spurious about it? This is the key blocker for me from scoring the paper higher and the only major criticism I have. In particular, A. the term was never formally defined in the paper. B. Are we looking at this as a property of the loss landscape (and hence a function of architecture), and that it is bad that such a point with very high loss could exist in the loss landscape? Or are we looking at this in the sense that it is bad because algorithms will usually find this point rather than "better" stationary points? C. I do not see a direct connection between "spurious" loss and "oversmoothing", in the sense that the connection to smoothness is not immediately clear to me

**Strengths And Weaknesses:**

Soundness: to the best of my knowledge, the math and derivation is generally sound (and quite elegant) except for the point that I raise in the limitation section that hopefully could be positively addressed. A small comment: at the first page, you mentioned P can be Laplacian etc. But afterwards, the entire paper revolved around P being stochastic, which the Laplacian is not. Perhaps refine the statement in the first page to avoid overloading/confusion around notation.

Presentation: The presentation is a major strength of the paper. Despite the technical nature of the material, I find the flow and writing clear and easy to follow.

Significance: The authors attempt to tackle a significant problem, and I believe their contributions are substantial.

Originality: to the best of my knowledge, this analysis (and perhaps even the problem formulation) is original.

---

> ### Author Rebuttal · Authors · 2026-03-30
>
> We thank the reviewer for their positive view of the paper and insightful remarks. We will take into account the minor points raised by the reviewer. We answer their major concern below.
>
> We agree that the term "spurious" was used rather loosely in the current version of the paper, thank you for raising this point that we did not foresee. Indeed, we used this designation to describe the stationary points due to oversmoothing that we identified, but since stationary points are not inherently bad, the term was rather informal. We then proved additional properties to argue that they are indeed "spurious": they can be provably bad in terms of loss, and do not happen at all for MLPs.
>
> Regarding point B., it is a bit a both: it *is* a specific property of the GNN architecture, since they do not happen for MLPs, and they *are* intuitively easy-to-reach, since they occur as soon as the last layer is trained (which we observe happens almost immediately in practice). Note that, as of yet, we have no rigorous proof that actual optimization algorithms will indeed reach these points first (but only intuition and empirical observation), this is an important outlook that would involve in-depth analysis of the *dynamics* of the optimization algorithm itself, a complex topic for deep NNs that seem to be out-of-reach for now. We will add this discussion.
>
> Nevertheless, we agree that the term "spurious" may indeed not be the best choice, we will clarify that this is an informal term, or maybe even remove it altogether in the final version.
>
> Regarding C., we will clarify the sketch of proof (which we believe provides more intuition than the theorem itself): these stationary points are indeed entirely due to forward and backward oversmoothing. As described in the proof, these phenomena create an upper bound on the gradients related to the *average output error*, that vanishes when the last layer is trained; and this would not happen if the forward signal was not "almost constant" for deep layers, or if the backward signal did not exploit almost-linear updates due to forward oversmoothing along with the limit $(P^\top)^k \to  \pi 1_n^\top$, at middle layers. But indeed the term "smoothing" provides little intuition by itself, we will clarify this discussion.

---

> > ### Author Rebuttal · Reviewer_nRS2 · 2026-04-04
> >
> > Thanks. I increase my score further.

---

### Decision · Program_Chairs · 2026-04-30

**Decision:**

Accept (regular)

**Comment:**

The paper studies "oversmoothing", a phenomenon in which due to averaging, the produced representations of a GNN do not capture much useful structure. They study the oversmoothing process in the "backward" (gradient update) step as well, and argue that due to this, GNNs can converge to "bad" stationary points.

Overall the reviews for the paper are quite positive, since it reveals some concrete issues in the training steps of common GNNs. The paper is of broad interest and has some clean insights. I recommend acceptance.